# FLAgellum Member 8 modulates extravascular distribution of African trypanosomes

Estefanía Calvo-Alvarez[1]*, Jean Marc Tsagmo Ngoune[1], Parul Sharma[1,2], Anneli Cooper[3], Aïssata Camara[4], Christelle Travaillé[1,5], Aline Crouzols[1], Annette MacLeod[3], Brice Rotureau[1,4]*

1 Trypanosome Transmission Group, Trypanosome Cell Biology Unit, INSERM U1201, Department of Parasites and Insect Vectors, Institut Pasteur, Université Paris Cité, Paris, France, 2 Sorbonne Université, ED515 Complexité du Vivant, Paris, France, 3 Wellcome Centre for Integrative Parasitology, College of Medical, Veterinary, and Life Sciences, Henry Wellcome Building for Comparative Medical Sciences, Glasgow, Scotland, United Kingdom, 4 Parasitology Unit, Institut Pasteur of Guinea, Conakry, Guinea, 5 Photonic BioImaging (UTechS PBI), Institut Pasteur, Université Paris Cité, Paris, France

* estefania.calvo@unimi.it (ECA); brice.rotureau@pasteur.fr (BR)

**Data Availability Statement:** All relevant data are within the paper and its Supporting Information files.

## Abstract

In the mammalian host, the biology of tissue-dwelling *Trypanosoma brucei* parasites is not completely understood, especially the mechanisms involved in their extravascular colonization. The trypanosome flagellum is an essential organelle in multiple aspects of the parasites' development. The flagellar protein termed FLAgellar Member 8 (FLAM8) acts as a docking platform for a pool of cyclic AMP response protein 3 (CARP3) that is involved in signaling. FLAM8 exhibits a stage-specific distribution suggesting specific functions in the mammalian and vector stages of the parasite. Analyses of knockdown and knockout trypanosomes in their mammalian forms demonstrated that FLAM8 is not essential *in vitro* for survival, growth, motility and stumpy differentiation. Functional investigations in experimental infections showed that *FLAM8*-deprived trypanosomes can establish and maintain an infection in the blood circulation and differentiate into insect transmissible forms. However, quantitative bioluminescence imaging and gene expression analysis revealed that *FLAM8*-null parasites exhibit a significantly impaired dissemination in the extravascular compartment, that is restored by the addition of a single rescue copy of *FLAM8*. *In vitro* trans-endothelial migration assays revealed significant defects in trypanosomes lacking *FLAM8*. FLAM8 is the first flagellar component shown to modulate *T. brucei* distribution in the host tissues, possibly through sensing functions, contributing to the maintenance of extravascular parasite populations in mammalian anatomical niches, especially in the skin.

## Author summary

*Trypanosoma brucei* parasites cause neglected tropical diseases termed human and animal African trypanosomiases. Transmitted by the bite of an infected tsetse fly, upon deposition in the skin of a mammalian host, these parasites occupy both the vasculature and

**Funding:** This work was supported by the Institut Pasteur. This work was supported by the French Government Investissement d'Avenir programme, Laboratoires d'Excellence ANR-10-LABX-62-IBEID (funding for ECA and JMTN) and ANR-11-LABX-0024-ParaFrap (funding for PS), and the French National Agency for Scientific Research via the projects ANR-14-CE14-0019-01 EnTrypa, ANR-18-CE15-0012 TrypaDerm, and ANR-19-CE15-0004-02 AdipoTryp granted to BR. ECA, JMTN, CT and PS were/are funded on these ANR grants. BR and AIC are funded by the Institut Pasteur and AiC by the Institut Pasteur of Guinea. AML and AnC are funded by a Wellcome Senior Fellowship to AML (209511/Z/17/Z). The funders had no role in study design, data collection and analysis, decision to publish, or preparation of the manuscript.

**Competing interests:** The authors have declared that no competing interests exist.

extravascular tissues. Currently, the biology of tissue-resident parasites is not well understood, and the parasite factors that mediate extravascular colonization are not known. Using quantitative *in vivo* bioluminescence imaging and *ex vivo* gene expression quantification in host infected tissues and blood, we reveal that the flagellar parasite protein FLAM8 modulates the extravascular dissemination of trypanosomes in the mammalian host. FLAM8 is known to act as a docking platform for signaling complexes in the flagellum, but we observe that it does not influence parasite differentiation into transmissible stages. However, we show that the absence of FLAM8 results in the loss of a key component of the flagellar adenylate cyclase signaling complexes, and reduces parasite migration through endothelial cell monolayers, suggesting that FLAM8 is important for parasite exchanges between the intravascular and the extravascular compartments. This work identifies a key trypanosome flagellar component involved in host-parasite interactions, including the modulation of parasite tropism and extravascular dissemination.

## Introduction

*Trypanosoma brucei* is an extracellular parasite responsible for African trypanosomiases, including sleeping sickness in humans and *nagana* in cattle. African trypanosomes are blood and tissue-dwelling protists transmitted by the bite of the blood-feeding tsetse fly (*Glossina* genus). In the mammalian host, parasites face different micro-environments, including deadly challenges by multiple types of host immune responses and variable availability of carbon sources. This requires major morphological and metabolic adaptations, driven by the activation of specific gene expression programs, that are critical for life-cycle progression [1–3]. Recently, the importance of extravascular tropism for *T. brucei* has been re-discovered in animal models: in addition to the brain, parasites occupy most mammalian tissues, especially the skin and the adipose tissues [4–7]. However, the role of extravasation and sequestration of trypanosomes in specific anatomical niches and the underlying molecular processes are not understood yet.

The trypanosome flagellum is an essential organelle anchored along the surface of the cell body and present in all stages of its development [8]. It is essential for parasite viability [9], cell division and morphogenesis [10], attachment to the tsetse salivary glands [11] and motility [12]. In the insect host, the flagellum remains at the forefront of the cell and is likely to be involved in sensory and signaling functions required for host-parasite interactions [13,14]. In the mammalian host, flagellar motility was shown to be critical for establishment and maintenance of bloodstream infection [15]. Nevertheless, the contributions of the trypanosome flagellum to parasite tropism and spatiotemporal dissemination outside vessels in the mammalian host remained to be explored.

A proteomic analysis of intact flagella purified from the insect stage of the parasite identified a group of flagellar membrane and matrix proteins with unique patterns and dynamics [16]. Amongst them, one large protein (3,075 amino acids) termed FLAgellar Member 8 (FLAM8) is present only at the distal tip of the flagellum in the insect procyclic form [16–18]. Interestingly, FLAM8 is redistributed along the entire length of the flagellum in mammalian forms, including in stumpy transmissible stages [19], which may imply a stage-specific function for this protein. Therefore, we hypothesized that FLAM8 could be involved in host-parasite interactions or in developmental morphogenesis. Here, we investigated the roles of FLAM8 in mammalian form parasites *in vitro* in terms of survival, proliferation, motility, and differentiation, as well as the *in vivo* dynamics of the intravascular and extravascular parasite

burden over the course of murine infection. Intriguingly, experimental infections in mice monitored and quantified by bioluminescence imaging and gene expression analyses demonstrated the involvement of FLAM8 in parasite dissemination in the extravascular compartments. In addition, *in vitro* transmigration studies detected impaired extravasation ability of *FLAM8*-deprived parasites, possibly resulting from the loss of components of flagellar adenylate cyclase signaling complexes.

## Results

### *FLAM8* RNAi silencing does not affect parasite survival

The differential distribution of FLAM8 in the flagellum of the different trypanosome stages [19] raises the question of its specific functions during the parasite life cycle. To investigate the potential role(s) of FLAM8 in the mammalian host, bloodstream form parasites were first engineered for inducible RNAi knockdown of *FLAM8* in a monomorphic strain expressing a mNeonGreen-tagged version of FLAM8. In order to monitor their behavior in mouse by whole-body imaging approaches, these *FLAM8*::*mNG FLAM8^RNAi* mutants were subsequently transformed to overexpress a chimeric triple reporter protein [20]. Upon RNAi induction with tetracycline for 72 h *in vitro*, *FLAM8* expression was reduced by 60% at the mRNA level (Fig 1A) and became undetectable at the protein level by immunofluorescence (Fig 1B). Parasite growth was monitored over 6 days upon induction of RNAi and no impact on proliferation was observed (Fig 1C), which suggests that FLAM8 is possibly not essential for survival of mammalian forms of the parasite in cell culture conditions.

Then, the linear correlation between the emitted bioluminescence and the number of parasites was assessed in an IVIS Spectrum imager prior to *in vivo* challenge (S1 Fig). To get insights into the function of FLAM8 in the mammalian host, groups of male BALB/c mice were infected by the intraperitoneal route with $10^5$ parasites of the parental, non-induced and induced cell lines (Fig 1D). *In vivo* RNAi silencing of *FLAM8* was maintained in mice by the addition of doxycycline in sugared drinking water 48 h prior infection and until the end of the experiment. The course of the infection was monitored daily by i) quantifying the parasitemia, and ii) acquiring the bioluminescent signal emitted by the parasites in the entire organism with an IVIS Spectrum imager. The number of parasites in the extravascular compartment at a given timepoint can be extrapolated by subtracting the known number of trypanosomes in the vascular system (parasitemia x blood volume, according to body weight) from the total number of parasites in the organism (total bioluminescence). No differences were detected, neither in the establishment of the infection and the subsequent variations in the number of intravascular parasites (Fig 1E, IV), nor in the number of parasites occupying extravascular tissues (Fig 1F, EV). Similar population profiles were observed over the course of the infection for both intra- and extravascular parasites in each of the three independent groups of infected mice (Fig 1D–1F). In the extravascular compartment, no significant differences were detected in terms of parasite dissemination over the entire animal body in any group (Fig 1G).

### *FLAM8* knockout does not affect trypanosome growth *in vitro*

Considering that i) *FLAM8* RNAi silencing efficiency was only partial (40% *FLAM8* mRNA left after 72 h of induction), ii) the efficacy of doxycycline-induced *FLAM8* repression could have been even lower *in vivo*, and that iii) the parental strain used for this first strategy was monomorphic (i.e. unable to differentiate into tsetse adapted stumpy stages), we reasoned that a gene knockout approach in a pleomorphic strain would be more appropriate to evaluate the potential role(s) of FLAM8 during the mammalian host infection. Therefore, a Δ*FLAM8* knockout cell line was generated in pleomorphic trypanosomes by homologous

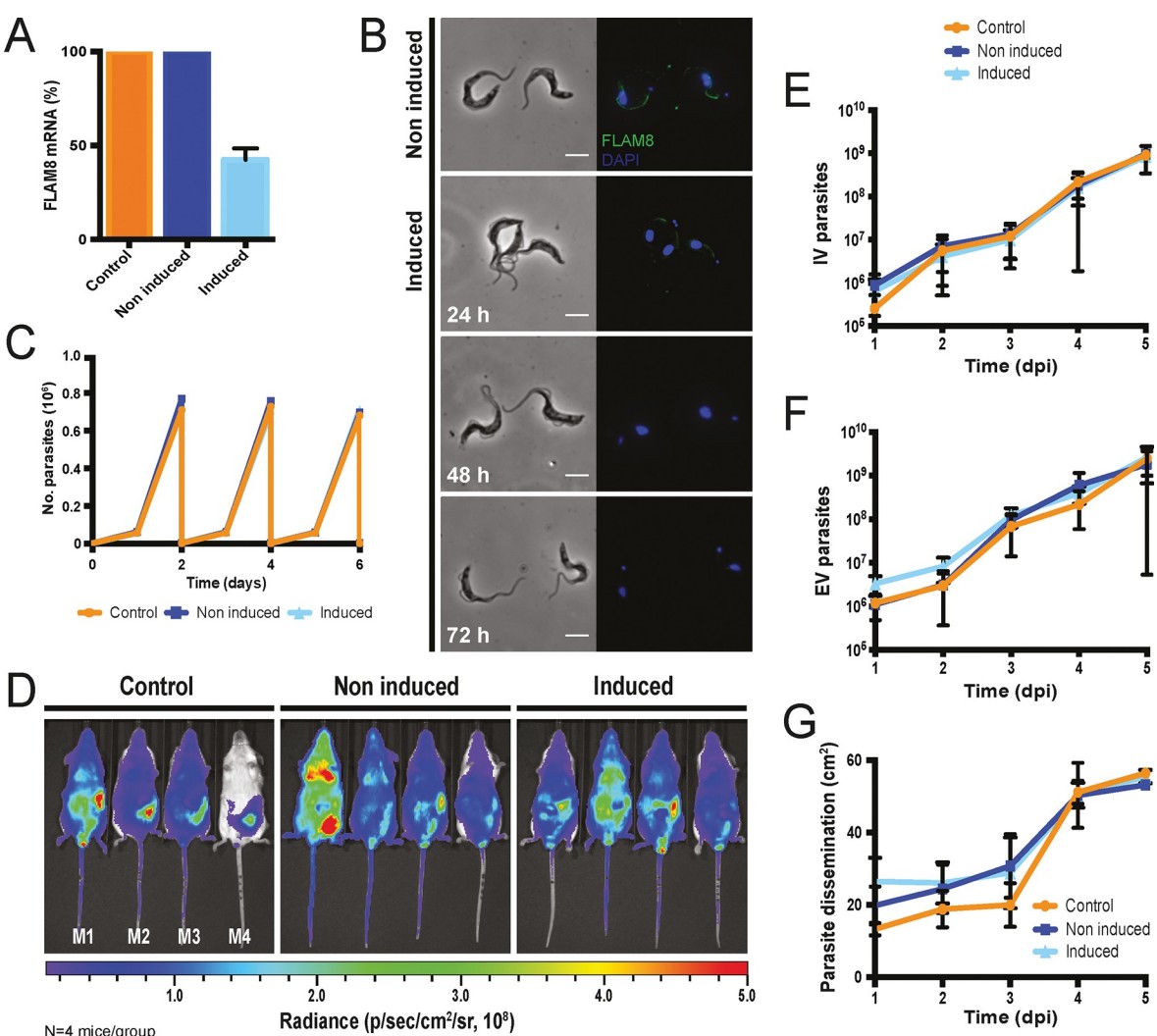

**Fig 1. Characterization of the *FLAM8::mNG FLAM8^RNAi* strain *in vitro* and *in vivo* functional investigation in the mammalian host.**
**A)** Expression of *FLAM8* mRNA assessed by RT-PCR by the comparative ΔΔC$_T$ method in control, non-induced and induced *FLAM8::mNG FLAM8^RNAi* parasites (72 h). **B)** Immunofluorescence pictures of non-induced (upper panels) and induced (bottom panels) *FLAM8::mNG FLAM8^RNAi* BSF during 72 h. Methanol-fixed trypanosomes were stained with an anti-mNG antibody (green) and DAPI for DNA content (blue). The scale bars represent 5 μm. **C)** Growth curves of control, non-induced and induced *FLAM8::mNG FLAM8^RNAi* BSF parasites. All cell lines received 1 μg tetracycline for 6 days. Control parasites are Lister 427 "Single Marker" BSF parasites that do not bear the pZJM-FLAM8 plasmid for RNAi silencing. Results represent the mean (± standard deviation, SD) of three independent experiments. **D)** Groups of 4 BALB/c mice were injected IP with either control, non-induced or induced *FLAM8^RNAi* BSF trypanosomes. One PBS-injected BALB/c animal was used as negative control. Representative normalized *in vivo* images of the bioluminescence radiance signal (in photons / second / cm$^2$ / steradian) emitted from BALB/c mice infected with control, non-induced and induced *FLAM8::mNG FLAM8^RNAi* parasites 4 days post-infection (non-infected technical control mice were negative for bioluminescence, not shown). RNAi silencing of *FLAM8* was maintained *in vivo* by the addition of doxycycline in sugared drinking water 48 h prior infection and until the end of the experiment. **E)** Number of parasites in the blood (intravascular, IV) of infected BALB/c mice during the course of the infection (5 days) counted from a tail bleed using a cytometer. **F)** Number of parasites in the extravascular compartment (extravascular, EV) of the same infected mice as in E). **G)** Dissemination of control, non-induced and induced *FLAM8::mNG FLAM8^RNAi* parasites, measured over the entire animal body (in cm$^2$) through the total bioluminescent surface, during the entire infection course. Results represent means ± standard deviation (SD).

recombination. The full replacement of one *FLAM8* allele and the partial replacement of the second one (to allow the use of a shorter *in situ* rescue sequence than the long *FLAM8* coding sequence) by distinct resistance cassettes was verified by whole-genome sequencing and PCR (S2 Fig). The absence of full or truncated FLAM8 protein expression was further assessed by

quantitative immunofluorescence analysis with an anti-FLAM8 antibody [19] targeting the region of FLAM8 that was not replaced in the second allele (Fig 2A). This was also confirmed at the mRNA level by RT-qPCR (S1 Table). The trypanosome cell lines generated were further transformed to express the chimeric triple reporter construct [20], and the linear correlation between the bioluminescence signal and the total number of parasites was analyzed for all strains both *in vitro* and *in vivo* (S3 Fig). In Δ*FLAM8* knockout parasites where a rescue copy of one *FLAM8* allele was added back into its endogenous locus, the FLAM8 distribution was restored along most of the flagellum length, as assessed by immunofluorescence analysis (Fig 2A). This was confirmed by quantifying the total fluorescence intensities of the FLAM8 signal along the full flagellum length in all parasite strains (Fig 2B): a background fluorescence signal was detected in knockout trypanosomes, while the rescue cell line presented an intermediate level of FLAM8 expression. No impact on parasite growth in culture conditions was observed (Fig 2C). Next, we investigated whether the loss of FLAM8 could have impacted the total length of the flagellum based on the measurement of the signal obtained with the axonemal marker mAb25: no difference was observed among BSF lines (Fig 2D). In addition, the absence of *FLAM8* did not affect parasite speed or linearity *in vitro*, neither in 0.5% methylcellulose liquid medium (Fig 2E and 2F) nor in 1.1% methylcellulose viscous medium (Fig 2G and 2H). Compatible with our previous observations after RNAi silencing, these results show that pleomorphic trypanosomes tolerated the loss of *FLAM8 in vitro*.

### *FLAM8* knockout affects trypanosome distribution in the mammalian host

Then, functional investigations in the mammalian host were performed by infecting BALB/c mice either with the pleomorphic parental strain, three distinct Δ*FLAM8* knockout subclones (resulting from independent recombination events) or one Δ*FLAM8* strain bearing a rescue copy of *FLAM8* (Figs 3 and S4). The infections were monitored daily for 4 weeks by quantifying the parasitemia and the bioluminescence signals emitted from whole animals (Fig 3A). Null mutant parasites were able to establish an infection in the bloodstream as well as in the extravascular compartment. Within the vasculature, the overall amounts of parasites throughout the whole experimental infection were slightly higher in mice infected with the three Δ*FLAM8* subclones as compared to mice infected with the parental strain, except between days 5 and 12 with a lower and variable parasitemia observed just after the first peak of parasitemia (Fig 3B and 3D). On the other hand, quantitative analyses of extravascular parasites showed a different scenario. Unlike in the intravascular compartment, significantly lower numbers of extravascular trypanosomes were observed between day 5 to 12, and from day 19 post-infection until the end of the experiment at day 27, evidencing an impaired extravascular colonization for Δ*FLAM8* parasites as compared to parental controls (Fig 3C and 3E). To note, distribution profiles of all pleomorphic strains were different from those observed in mice infected with monomorphic parasites: larger amounts of trypanosomes were seen occupying the extravascular compartment reaching up to $5-8\times10^9$ parasites, while the maximum amount found in the bloodstream never exceeded $7\times10^7$ trypanosomes (Fig 3C and 3B, respectively).

Furthermore, quantification of the parasite spreading over the whole animal bodies showed that the depletion of *FLAM8* resulted in significantly impaired dissemination of Δ*FLAM8* null mutant parasites in the extravascular compartment (days 5 to 14, 19 and 27 post-infection). This was mostly restored in trypanosomes bearing a rescue copy of *FLAM8* (Fig 3F).

### *FLAM8* knockout impairs extravascular trypanosome dissemination

We reasoned that the accuracy of the intra- Vs. extravascular parasite population estimation from bioluminescence detection on small regions of interest could be limited. Therefore, the

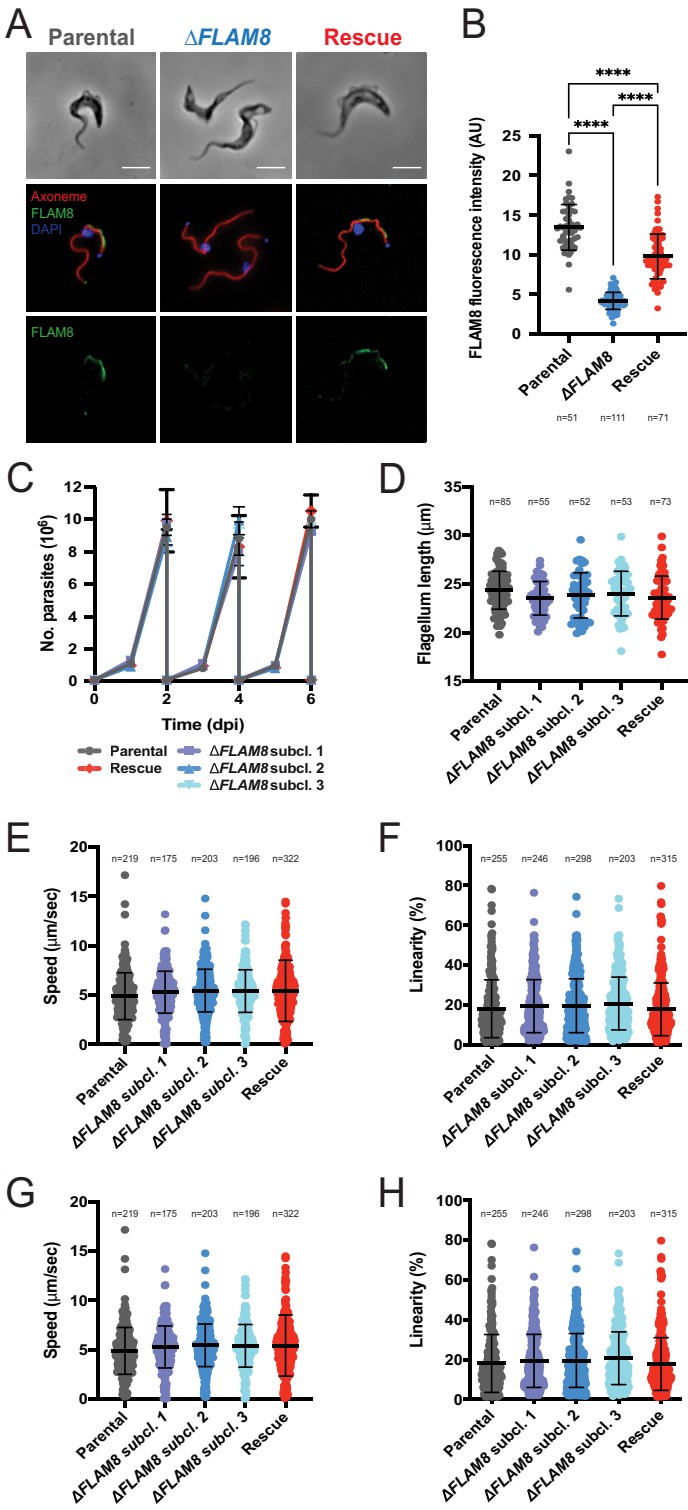

**Fig 2. Characterization of ΔFLAM8 null mutants *in vitro*. A)** Immunofluorescence pictures of parental, ΔFLAM8 knockout and rescue pleomorphic BSF parasites labelled with the anti-FLAM8 (green) and mAb25 (axoneme in red) antibodies, DAPI staining for DNA content (blue). Scale bars show 5 μm. **B)** Quantification of the total fluorescence intensity of FLAM8 along the entire flagellum length in methanol-fixed parental, ΔFLAM8 knockout and rescue parasites. **C)** Growth curve of one parental, three ΔFLAM8 subclones and one rescue pleomorphic BSF trypanosome

cell lines over 6 consecutive days. **D)** Measurements of the flagellum length based on the axonemal marker mAb25 profiles in parental, three Δ*FLAM8* subclones and rescue parasites. No statistical differences were found. **E-H)** Motility tracking analysis showing the average speeds (E and G) and linearity (F and H) of BSF cell lines in matrix-dependent culture medium containing 0.5% (E-F) or 1.1% (G-H) methylcellulose. No statistical differences were observed. The number of parasites considered for quantifications (N) is indicated under and above graphs in (B) and (D-H), respectively. Results represent the mean ± standard deviation (SD) of three independent experiments. Statistical tests included one-way ANOVA and Tukey's ad-hoc post-tests for multiple comparisons.

same experiment was repeated, with a lower time resolution, yet with comparable trends in terms of extravascular parasite populations and parasite dissemination according to the strain (Figs 4A–4C and S5). At the end of this second experimental challenge (Day 24), most trypanosomes were removed from the vascular system by saline perfusion prior to organ dissection. Individual organs were then collected for quantifying the expression of different parasite genes by RT-qPCR. The total number of parasites in each sample was calculated per mg of tissues by using a *Tubulin* RT-qPCR standard curve. To better compare the variations of the parasite populations in each compartment between strains, the Delta number of parasites was calculated as the difference between the number of parasites in each tissue sample of a given mouse and the number of parasites in the blood sample from the same mouse.

First, the absence of *FLAM8* transcripts was confirmed in animals infected with Δ*FLAM8* parasites, whereas *FLAM8* mRNAs were detected in mice infected with both the parental and rescue strains (S1 Table). Normalized *Tubulin* expression was then used to compare the parasite densities between organs and strains (Figs 4D, 4E and S6). In all mice, the highest number of extravascular parasites was observed in the skin. In all organs, a marked decrease in the number extravascular parasites was detected from mice infected with Δ*FLAM8* parasites as compared to mice infected by parental and rescue parasites (Figs 4D and S6). When considering the average difference between the number of parasites in organs versus blood by strains, the statistically significant decrease in extravascular *FLAM8*-deprived parasite densities appeared even more clearly (Fig 4E). The rescue cell line recapitulated EV parasite profiles in all organs examined with an average Delta number of parasites that was not statistically different from that of the parental line.

We reasoned that this reduced extravascular dissemination of *FLAM8*-deprived parasites could possibly be due to: (1) a lower proliferation rate, (2) a defect in motility, (3) a higher rate of differentiation into non-proliferative stumpy forms, and / or (4) an extravasation defect. The two first hypotheses could immediately be discarded as no difference was observed between strains, neither in cell growth *in vitro* (Fig 2C) and *in vivo* (Figs 3B and 4A), nor in cell motility *in vitro* (Fig 2E–2H). The two last hypotheses were then successively tested.

## The absence of FLAM8 does not affect parasite differentiation

The systemic reduction in the extravascular Δ*FLAM8* parasite population could result from a disequilibrium in the parasite differentiation into non-proliferative transmissible forms. Therefore, we assessed whether the absence of FLAM8 could have impacted the differentiation of proliferative slender into transmissible stumpy forms in the blood and organs by immuno-fluorescence or RT-qPCR. In cultured parasites, the absence of *FLAM8* did not significantly alter differentiation (Fig 5A and 5B), and freshly differentiated *FLAM8*-deprived stumpy parasites were able to further differentiate and maintain *in vitro* as procyclic trypanosomes, as were the parental and rescue strains (Fig 5C and 5D). *In vivo*, the natural differentiation of proliferative slender into non-proliferative stumpy parasites was confirmed by RT-qPCR on blood and dissected organs at the end of the experiment (Fig 5E, Day 24 of the second *in vivo* challenge). For each sample, *Tubulin* expression was used to normalize the *PAD1* mRNA levels to

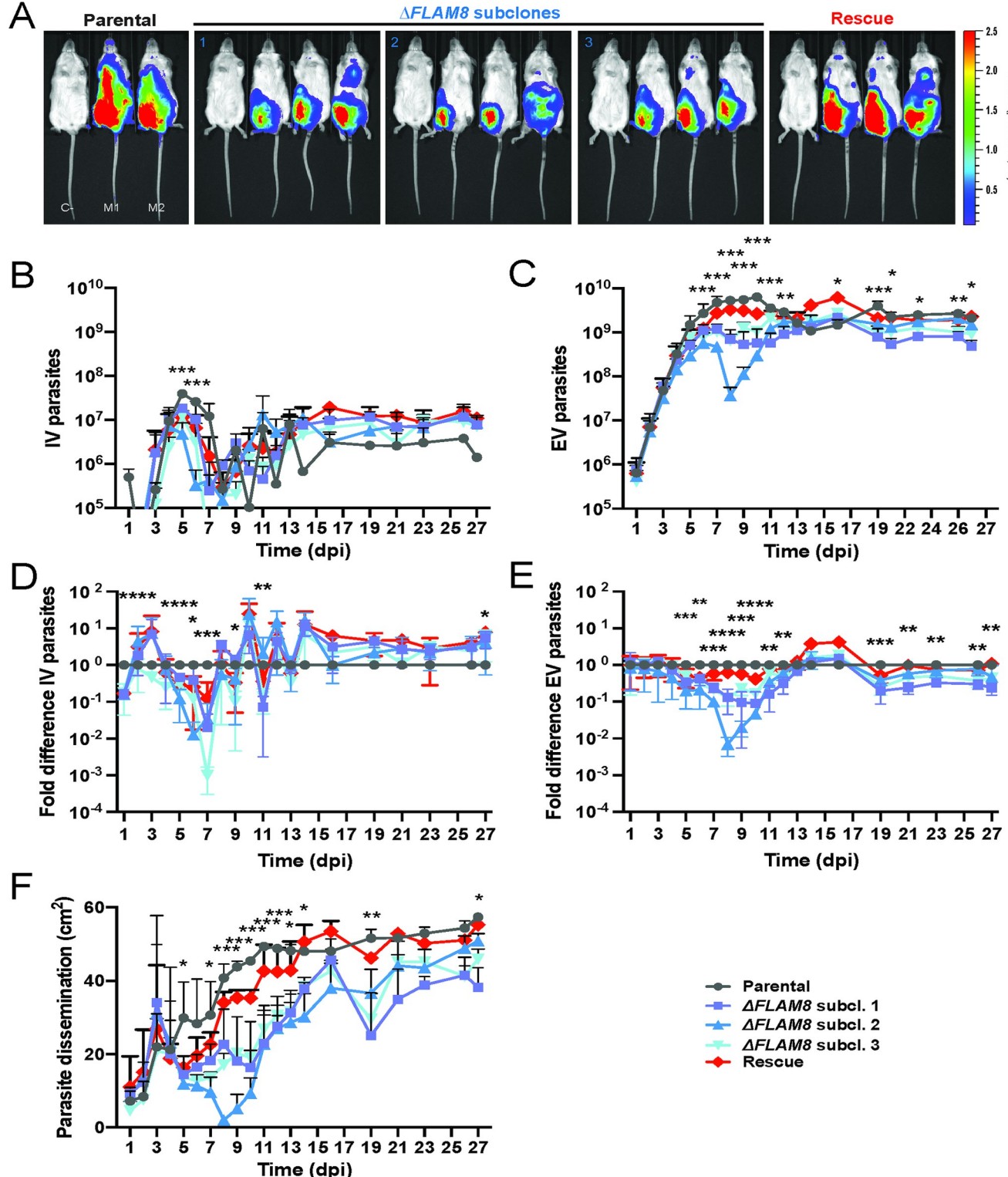

**Fig 3. Functional investigations on the Δ*FLAM8* null mutants *in vivo* in the mammalian host.** Groups of 3 BALB/c mice were injected IP with either one parental, three Δ*FLAM8* null subclones or one rescue strains. One PBS-injected BALB/c animal was used as negative control. **A)** Normalized *in vivo* images of the bioluminescence radiance intensity (in photons / second / cm² / steradian) emitted 8 days post-infection in BALB/c mice infected with parental, three Δ*FLAM8* subclones or rescue parasites (non-infected control mice C- were negative for bioluminescence). **B)** Total number of parasites in the blood of infected mice (intravascular, IV) daily counted from tail bleeds using a cytometer over 4 weeks. Statistically significant differences (p<0.01)

are indicated with one, two or three asterisks (*, **, ***) representing differences between the parental strain and one, two or three Δ*FLAM8* subclones, respectively. **C)** Total number of extravascular (EV) trypanosomes in the same mice. Statistically significant differences between the parental strain and Δ*FLAM8* subclones are indicated as in B). **D-E)** Variations of the total numbers of intravascular (IV as in B) and extravascular (EV as in C) parasite populations in the same mice plotted as fold differences to values obtained with mice infected with parental trypanosomes at the same time point. **F)** Dissemination of the parental, three Δ*FLAM8* subclones and rescue parasite strains, measured over the entire animal body (in cm$^2$) through the total surface of bioluminescent signal, during the entire infection course. Statistically significant differences between the parental strain and Δ*FLAM8* subclones are indicated as described above. Results represent means ± standard deviation (SD). Statistical tests included two-way ANOVA and Tukey's ad-hoc post-tests for multiple comparisons in B-F. Detailed individual data are provided in S4 Fig.

compare the average levels of parasites expressing *PAD1* mRNAs over the total parasite populations between organs and strains (Fig 5E), with a higher Delta CqPAD1-CqTub correlating with a lower amount of *PAD1* transcripts in the organ. *PAD1* transcripts were detected in each organ at least in one individual, showing that all tested organs represent a suitable environment for the parasites to differentiate into transmissible forms (S1 Table). In mice infected with the parental and rescue strains, the highest levels of *PAD1* transcripts were detected in blood and skin, suggesting an accumulation or an increased rate of differentiation in organs directly involved in parasite transmission (S1 Table). No *PAD1* transcripts were detected in the gut, liver and spleen in mice infected with the parental and rescue strains, whereas *PAD1* transcripts were detected in these three organs from mice infected with the Δ*FLAM8* strains. However, when considering the average Delta CqPAD1-CqTub in the entire organisms by strains, the relative proportions of parasites expressing *PAD1* mRNAs were not significantly different among groups, confirming that FLAM8 is not involved in parasite differentiation (Fig 5E). In addition, an anti-PAD1 immunofluorescence staining of blood smears sampled at the first peak of parasitemia visually confirmed the presence of differentiated stumpy parasites in all infected animal groups (Fig 5F, Day 5 of the first *in vivo* challenge).

## Parasite extravasation is impaired in *FLAM8*-deprived trypanosomes

To test the last hypothesis, we asked whether the absence of FLAM8 might influence the way parasites traverse the vessel walls to access extravascular tissues. We first established a transmigration assay using human umbilical vein endothelial cells (HUVECs) grown to confluence on polyester transwell inserts with 3 μM pores, separating two chambers to mimic the vascular endothelium *in vitro* (Fig 6A). Once HUVECs have reached confluence, 10$^6$ parental, *FLAM8*-deprived or rescue trypanosomes were added to the upper chamber and incubated for 24h. The number of trypanosomes that migrated through the endothelial monolayer into the lower chamber and the number of non-migrating trypanosomes remaining in the upper chamber of the transwell system were counted by flow cytometry to determine the transmigration percentage. Interestingly, all *FLAM8*-knockout subclones showed a significant reduction in transmigration relative to parental controls (25,6%, 70,7% and 33,8% of parasites crossing for KO1, 2 and 3, respectively) (Fig 6B). Rescue trypanosomes expressing only one *FLAM8* allele exhibited an intermediate transmigration phenotype in which 59,2% of the parasites migrated through the endothelial monolayer (Fig 6B), which was lower than for parental trypanosomes but significantly higher than for KO1 and KO3 null mutants (Fig 6B). This demonstrates a strong impairment of parasites lacking *FLAM8* to cross through a confluent layer of endothelial cells, which was partially restored in rescue parasites.

Knowing that, in mammalian forms, FLAM8 was recently identified to be part of a flagellar complex including the cyclic AMP response protein 3 (CARP3) [21], we reasoned that this trans-endothelial crossing impairment could be caused by a defect in sensing and / or signaling. To assess how the absence of FLAM8 was impacting CARP3 localization and/or abundance, CARP3 distribution was investigated by quantitative immunofluorescence in all strains

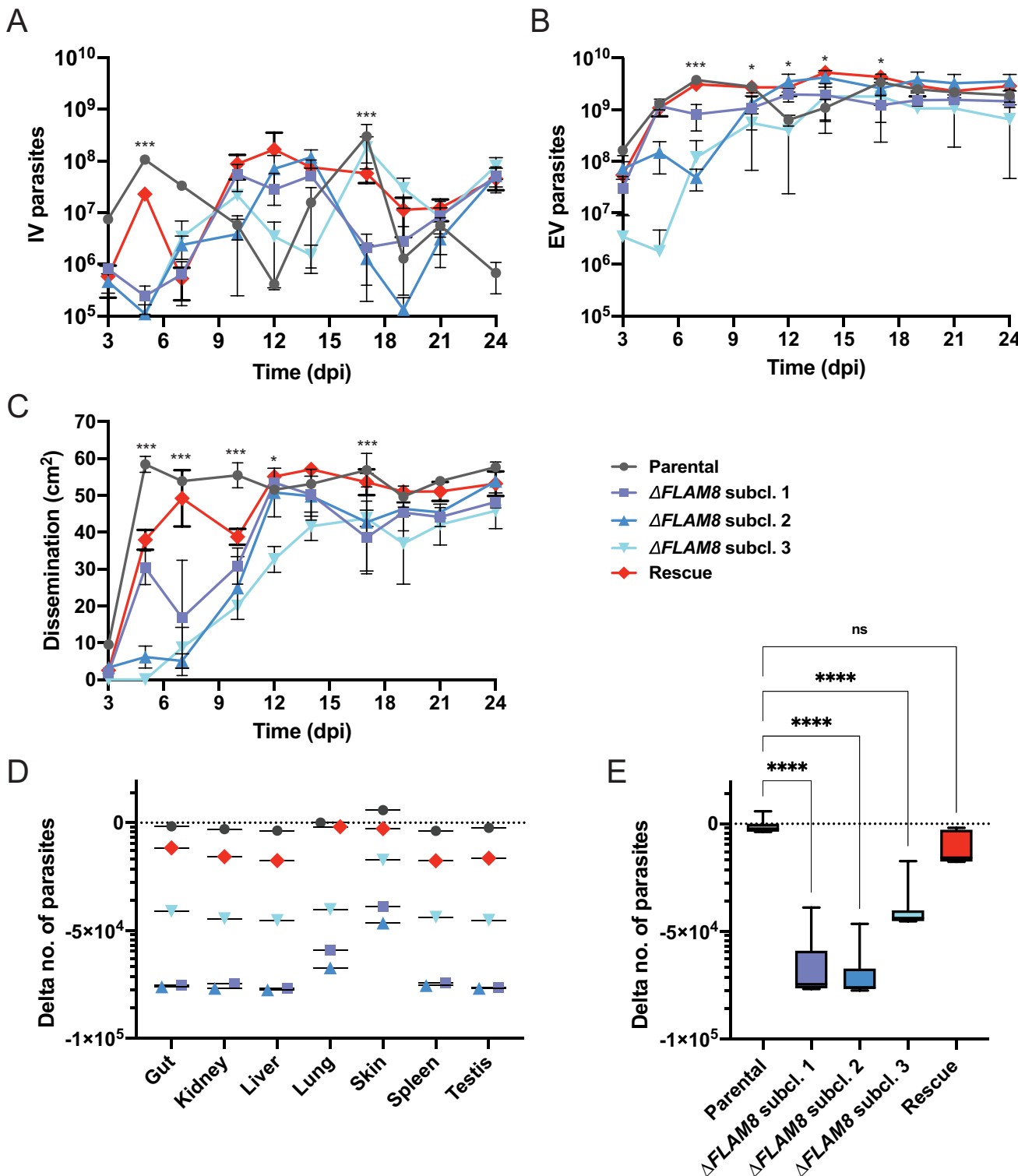

**Fig 4. The absence of *FLAM8* reduces extravascular trypanosome dissemination.** In a second experimental infection, groups of 3 BALB/c animals were injected IP with either parental, three Δ*FLAM8* null subclones or rescue strains, while one PBS-injected BALB/c animal was used as negative control. At day 24 post-infection, animals were euthanized, and the gut, kidney, liver, lung, skin, spleen and testes were collected to quantify the parasite proportion in each organ. **A)** Total number of intravascular parasites (IV) during the infection period. **B)** Total number of extravascular (EV) trypanosomes in the same mice. **C)** Dissemination of the parental, three Δ*FLAM8* subclones and rescue parasite strains, measured over the entire animal body (in cm$^2$) through the total surface of

bioluminescent signal, during the entire infection course. Statistically significant differences (p<0.01) are indicated with one or three asterisks (*, ***) representing differences between the parental strain and one or three ΔFLAM8 subclones, respectively. Detailed individual data are provided in S5 Fig **D)** Delta number of parasites per dissected organs and strains. The total number of parasites in each sample was calculated per mg of tissues by using a *Tubulin* RT-qPCR standard curve. To better compare the variations of the parasite populations in each compartment between strains, the Delta number of parasites was calculated as the difference between the number of parasites in each tissue sample of a given mouse and the number of parasites in the blood sample from the same mouse. **E)** Delta number of parasites per strain calculated as the average of values for individual organs shown in D. Statistical differences (p<0.0001) according to one-way ANOVA and Dunnett's comparison tests. Detailed individual data are provided in S1 Table and S6 Fig.

upon methanol fixation (Fig 6C and 6D). As expected, CARP3 was detected along the entire flagellum length in parental mammalian forms, but not in ΔFLAM8 parasites where the fluorescent signal was weaker and restricted to the distal part of the flagellum (Fig 6C and 6D). This demonstrates the presence of at least two distinct pools of CARP3, one being independent of FLAM8 for its localization to the anterior part of the flagellum. The presence of a *FLAM8* rescue copy in ΔFLAM8 parasites restored the detection of CARP3 along the entire flagellum, yet in lower amounts than in parental cells (Fig 6C and 6D).

In total, these data show that FLAM8 is likely involved in a cellular pathway modulating trypanosome extravasation and consequently trypanosome dissemination in the extravascular compartment of the mammalian host.

## Discussion

The differential localization of FLAM8 from the very distal tip in tsetse midgut procyclic parasites to the entire flagellum length in the mammalian-infectious stages [19] prompted us to speculate that FLAM8 could play a distinct and specific role in each host. Here, we present for the first time the connection of a flagellar protein with the efficiency of trypanosomes to disseminate outside the mammalian host vasculature, especially in the skin. Quantitative analyses of experimental animal infections monitored by bioluminescence imaging and gene expression analysis showed that the absence of FLAM8 impairs parasite extravasation and dissemination in the host extravascular compartment over the time of the infection, which was mostly recovered by the integration of a single rescue copy of the *FLAM8* gene in the endogenous locus.

### 1. Inter- and intra-individual variations

Inter-individual heterogeneity is common during experimental *in vivo* infections. Nevertheless, the overall parasite population dynamics in the 3 groups of mice infected with each of the 3 ΔFLAM8 sub-clones (18 mice in total) followed very comparable trends in the two main experiments (Figs 3 and 4). Both data sets showed that ΔFLAM8 parasites can colonize the extravascular compartments, yet in densities that remained below those of the parental and rescue populations over most of the infection course. This was confirmed by RT-qPCR analyses at the end of the second experiment (Fig 4D and 4E).

At the individual level, the overall parasite distribution in each compartment of the host (different vascular sub-compartments, organs and tissues) evolves differently and in a dynamic manner over the course of an infection. In addition, within each compartment, the parasite distribution is not homogenous. Here, for instance, this was reflected by the variable dissemination of parasites in the skin. Combining parasite quantification by whole-organism bioluminescence imaging and RT-qPCR in specific organs also demonstrated that the size of the extravascular parasite population in each organ was differentially impacting the total extravascular population, with the skin representing the most important anatomical niche among all tested organs.

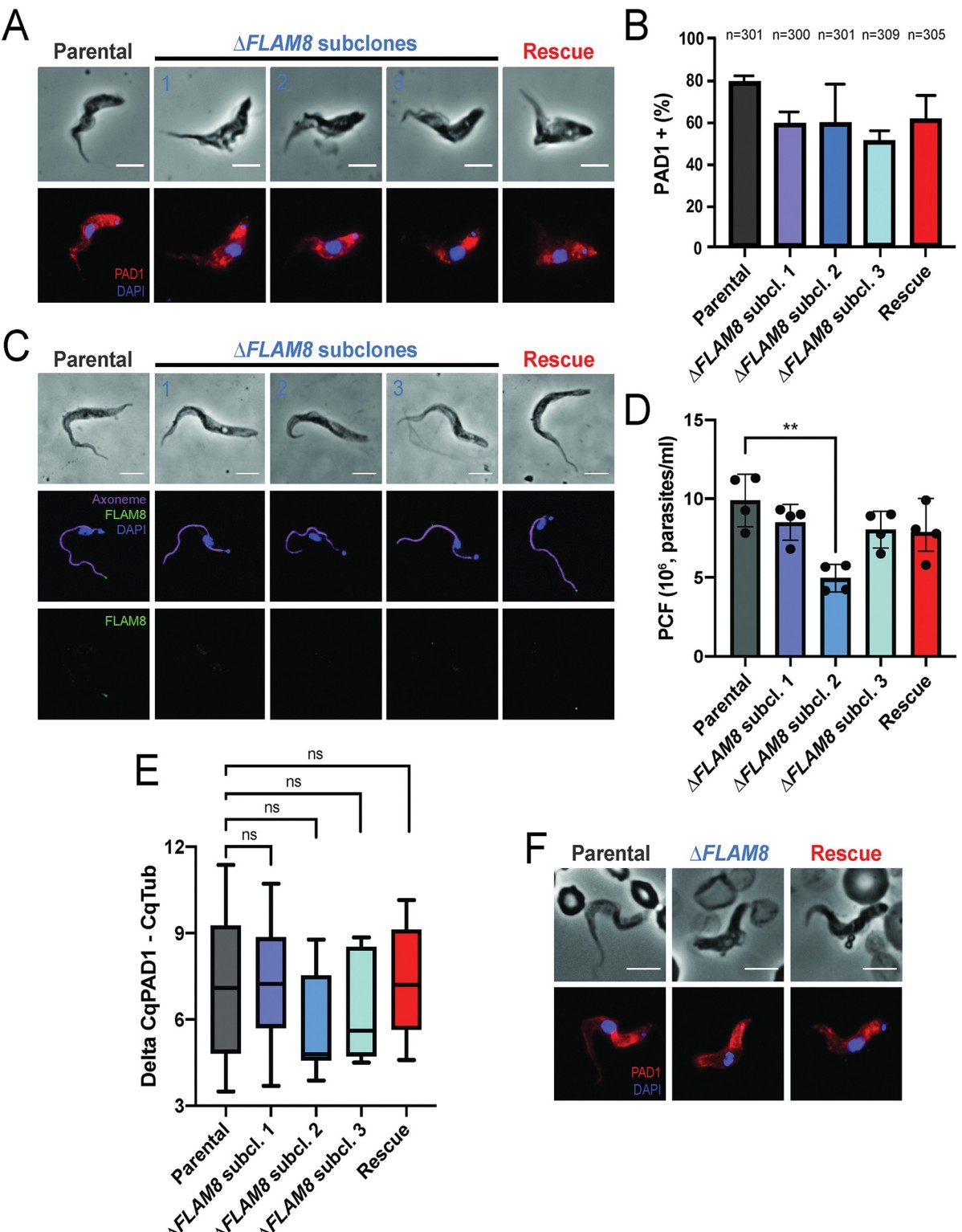

**Fig 5. Parasite differentiation is not impacted in *FLAM8*-depleted trypanosomes. A)** Representative immunofluorescence pictures of stumpy parasites after *in vitro* differentiation from proliferative slenders of parental (left panel), three Δ*FLAM8* subclones (middle panels) and rescue (right panel) parasite strains upon *in vitro* treatment with a nucleotide 5'-AMP analog. Methanol-fixed parasites were labelled with the anti-PAD1 antibody (red), and DAPI staining for DNA content (blue). Scale bars show 5 μm. **B)** Quantification of the proportion of stumpy trypanosomes in all pleomorphic cell lines after *in vitro* differentiation. The number of parasites considered for quantification (n) is

indicated above the graph. No significant differences were found (one-way ANOVA and Tukey's comparisons test). Results represent the mean ± standard deviation (SD) of three independent experiments. **C)** Selected immunofluorescence images of freshly *in vitro* differentiated procyclic cells. Trypanosomes were doubly labelled with anti-FLAM8 (green) and mAb25 (axoneme in magenta), DAPI staining showing DNA content in blue. Scale bars show 5 µm. **D)** Upon differentiation, procyclic forms (PCF) of parental, three Δ*FLAM8* subclones and rescue parasite strains were equally diluted, and their *in vitro* growth assessed. Statistical differences (p<0.01) were observed only when comparing parental and KO 2 PCF trypanosomes (one-way ANOVA and Tukey's comparison test). Results represent the mean ± standard deviation (SD) of four independent experiments. **E)** Average relative proportions of *PAD1* expression levels quantified by RT-qPCR on blood and dissected organs at day 24 of the second *in vivo* challenge. *Tubulin* expression was used to normalize the *PAD1* mRNA levels in all parental-, Δ*FLAM8*- and rescue-infected mice samples. **F)** Representative immunofluorescence images of naturally differentiated stumpy trypanosomes from parental, one selected Δ*FLAM8* subclone and rescue strains isolated from mouse blood during the first peak of parasitemia (first experimental *in vivo* infection). Parasites were labelled with anti-PAD1 antibody (red), and DAPI staining for DNA content (blue). Scale bars show 5 µm.

Observing a variable virulence among clones of the same pleomorphic strain is frequent, hence the importance of testing different parasite clones. Here, all three Δ*FLAM8* sub-clones shown a significantly lower transmigration efficiency as compared to their parental strain, although clone 2 presented a higher transmigration efficiency as compared to the other two. Moreover, due to the restoration of a single allele of the *FLAM8* gene, add-back trypanosomes only presented an 'intermediate' rescue phenotype in all experiments performed.

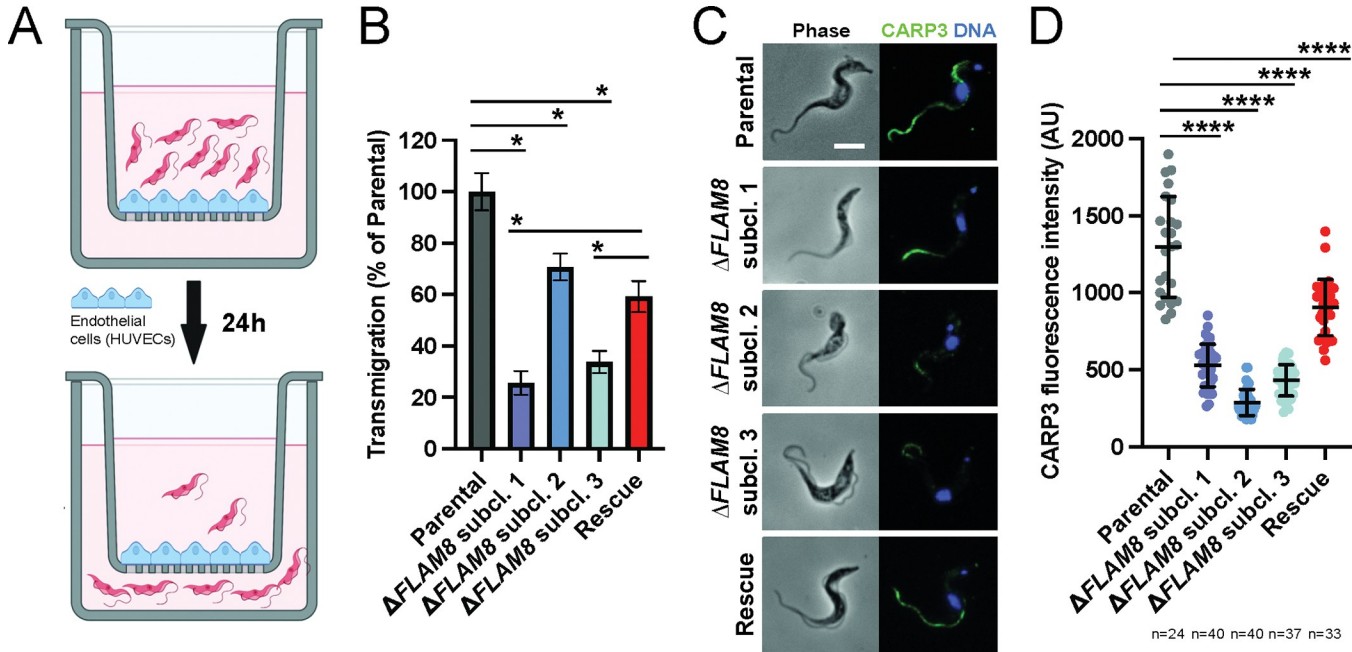

**Fig 6. Parasites lacking *FLAM8* are impaired in transmigrating through endothelial cells *in vitro*. A)** Schematic representation of the trans-endothelial migration assay, showing the transwell system (Boyden chamber) containing the upper and the lower compartments, the monolayer of endothelial HUVEC cells and slender trypanosomes seeded on the top of the chamber for 24 hours. **B)** After this period, parasites within the upper and lower chambers were counted and values further used to calculate the proportion of parental, three Δ*FLAM8* subclones and rescue parasites that migrated through the endothelial monolayer into the lower compartment. Trypanosome transmigration was compared between the FLAM8 mutant cell lines and the parental reference line (100% of trans-endothelial migration) using the Generalised Linear Model function in R with a Gaussian family function and proportion of transmigration as the dependent variable. A probability value of p<0.001 was considered significant (*). Error bars show SD. **C)** Representative immunofluorescence images of cultured slender trypanosomes from parental, Δ*FLAM8* subclone and rescue strains. Methanol-fixed parasites were labelled with anti-CARP3 antibody (green intensity normalized), and DAPI staining for DNA content (blue). Scale bars show 5 µm. **D)** Quantification of CARP3 fluorescence signal in normalized images from methanol-fixed parental, Δ*FLAM8* subclones and rescue trypanosomes, expressed as the ratio between the fluorescent CARP3 signal intensity along the entire flagellum divided by the length of individual flagella. The number of parasites considered for quantification (n) is indicated under the graph. Statistical differences (p<0.0001) according to one-way ANOVA and Dunnett's comparison test. Results represent the mean ± standard deviation (SD) of two independent experiments.

## 2. FLAM8 and trypanosome transmission

In the bloodstream, the balance between proliferative slender parasites and tsetse-adapted stumpy forms responds to a quorum sensing mechanism involving the production of oligo-peptides and their reception through a specific transporter [22]. The absence of FLAM8 did not alter the ability of the parasites to differentiate into transmissible stumpy forms at the two observed time points (first peak of parasitemia and 3 weeks after infection). Although stumpy proportions were not evaluated over the entire course of the infection, this evidence suggests that FLAM8 does not play an integral role in this process. In addition, the retained ability of *FLAM8*-deprived stumpy cells to differentiate into procyclic forms *in vitro* suggests that they could further pursue their cyclical development upon ingestion by a tsetse fly.

Extravascular trypanosomes occupy the interstitial space of several organs, including the central nervous system, testes, adipose tissues, and skin [4–7, 23–25]. The relevance of skin-dwelling trypanosomes in parasite transmission has been demonstrated by xenodiagnosis experiments, early after the infective bite [26], or later in the infection [4], even in the absence of detectable parasitemia. More recently, the presence of extravascular trypanosomes was confirmed in the skin of confirmed and suspected cases of sleeping sickness [27]. Here, we show that all tested organs represent a suitable environment for the parasites to differentiate into transmissible forms, yet the highest amounts of *PAD1* transcripts were detected in blood and skin, suggesting an increased rate of differentiation, or an accumulation of differentiated stumpy forms in organs directly involved in parasite transmission. In total, the impaired spreading of *FLAM8*-null parasites over the extravascular compartment would mathematically reduce the probability for parasites to be ingested by tsetse flies. This could be especially significant for the skin, that has the highest parasite density, and the highest overall parasite load due to its size, at the direct interface with insect vectors.

## 3. On the possible cellular function(s) of FLAM8

Proliferative slender trypanosomes are highly mobile [28] and this motility was proved to influence virulence *in vivo*. For instance, the lack of propulsive motility in flagellar dynein light chain 1 (*LC1*) knockout mutants resulted in the inability of trypanosomes to establish an infection in the bloodstream [15]. Here, *FLAM8* depletion in pleomorphic mammalian forms did not alter parasite growth and cell motility neither in liquid medium, nor in matrix-containing medium with a viscosity closer to the parasite micro-environment *in vivo*. However, quantitative analyses showed that *FLAM8*-null trypanosomes were less numerous in extravascular tissues as compared to parental controls. Assuming that parasite motility could be different in tissues and interstitial spaces with biophysical properties distinct from those in the blood [28, 29], one cannot exclude that the motility of ΔFLAM8 knockout parasites might be somehow altered in the extravascular compartment. Intravital imaging for motility analyses at the cell level would be needed to confirm this hypothesis.

Historically, most studies on *T. brucei* virulence in experimental infections have considered the blood circulation almost as the sole host compartment parasitized by trypanosomes, whereas extravascular parasite niches and the underlying exchanges between both compartments have been underestimated for long. De Niz and colleagues recently identified adhesion molecules as key players for tissue tropism [30]. They showed that reservoir establishment happens before vascular permeability is compromised, suggesting that extravasation is an active mechanism, and depends on trypanosome interactions with endothelial surface adhesion molecules, such as E-selectin, P-selectins, or ICAM2 [30]. Here, we showed that the absence of FLAM8 led to a strong impairment in parasite extravasation. The fact that *FLAM8*-null trypanosomes were not able to disseminate properly over the extravascular compartment could

somehow possibly imply defects in the way parasites sense their microenvironment or interact with the host endothelial cells, resulting in an alteration of their extravascular tropism.

The insect forms´ coordinated social motility *in vitro* is linked to cAMP signaling at the flagellar tip [31, 32], i. e. where FLAM8 localizes [19]. The architecture of an adenylate cyclase complex in the tip nanodomain was recently shown to be essential for social motility and salivary gland colonization [21]. In this complex, CARP3 interacts with the catalytic domain of adenylate cyclases and regulates abundance of multiple adenylate cyclase isoforms. We recently demonstrated that the CARP3 tip localization depends on the presence of FLAM8 acting as a scaffold protein [21]. Thus, trypanosome migration and transmission in the tsetse vector specifically depend on adenylate cyclase complex-mediated signaling in the tip nanodomain including FLAM8 [21]. Interestingly, we have recently shown that CARP3 and FLAM8 are both redistributed along the entire length of the flagellum during their differentiation to the mammalian stage, and that they further remain associated in flagellar complexes in mammalian forms [21]. Here, this was confirmed by the significant decrease of the pool of CARP3 detected in the proximal flagellum region of *FLAM8*-deprived parasites. Altogether, these data could reflect a possible specific function of some CARP3-containing signaling complexes depending on FLAM8 for their sub-flagellar localization. Environmental sensing and / or signaling, possibly through direct contact with host cell receptors may play a role in extravasation. We hypothesize that the absence of FLAM8 would destabilize or delocalize these signaling complexes, impairing parasite sensing, signaling and / or adhesion functions, hence preventing the parasite to efficiently cross vascular endothelia.

To our knowledge, FLAM8 is the first flagellar component affecting parasite extravasation and their subsequent dissemination in the extravascular host tissues *in vivo*. Further investigations on the FLAM8 interactions with other possible partners in the flagellum would help to unravel the roles of this fascinating and essential organelle, especially regarding the modulation of trypanosome tropism, extravasation and spreading in their mammalian hosts, and its implications in parasite virulence and transmission.

## Materials and methods

### Ethics statement

This study was conducted in strict accordance with the recommendations from the Guide for the Care and Use of Laboratory Animals of the European Union (European Directive 2010/63/UE) and the French Government. The protocol was approved by the "Comité d'éthique en expérimentation animale de l'Institut Pasteur" CETEA 89 (Permit number: 2012–0043 and 2016–0017) and undertaken in compliance with the Institut Pasteur Biosafety Committee (protocol CHSCT 12.131).

### Strains, culture and *in vitro* differentiation

The AnTat 1.1E Paris pleomorphic clone of *Trypanosoma brucei brucei* was derived from a strain originally isolated from a bushbuck in Uganda in 1966 [33]. The monomorphic *T. brucei* strain Lister 427 [34] was also used. All bloodstream forms (BSF) were cultivated in HMI-11 medium supplemented with 10% (v/v) fetal bovine serum (FBS) [35] at 37˚C in 5% $CO_2$. For *in vitro* slender to stumpy BSF differentiation, we used 8-pCPT-2′-O-Me-5′-AMP, a nucleotide analog of 5'-AMP from BIOLOG Life Science Institute (Germany). Briefly, $2x10^6$ cultured pleomorphic AnTat 1.1E slender forms were incubated with 8-pCPT-2′-O-Me-5′-AMP (5 μM) for 48 h [36]. For specific experiments, *in vitro* differentiation of BSF into procyclic forms was performed by transferring freshly differentiated short stumpy forms into SDM-79 medium supplemented with 10% (v/v) FBS, 6 mM cis-aconitate and 20 mM glycerol at 27˚C [37].

Monomorphic BSF "Single Marker" (SM) trypanosomes are derivatives of the Lister 427 strain, antigenic type MITat 1.2, clone 221a [38], and express the T7 RNA polymerase and tetracycline repressor. *FLAM8^RNAi* cells express complementary single-stranded RNA corresponding to a fragment of the *FLAM8* gene from two tetracycline-inducible T7 promoters facing each other in the pZJM vector [39] integrated in the rDNA locus [40]. Addition of tetracycline (1 μg/mL) to the medium induces expression of sense and anti-sense RNA strands that can anneal to form double-stranded RNA (dsRNA) and trigger RNAi. For *in vivo* RNAi studies in mice, doxycycline hyclate (Sigma Aldrich) was added in sugared drinking water (0.2 g/L doxycycline hyclate combined with 50 g/L sucrose).

## Generation of *FLAM8* RNAi mutants

For the generation of the *FLAM8^RNAi* cell lines, a 380 bp (nucleotides 6665–7044) fragment of *FLAM8* (Tb927.2.5760), flanked by 5' HindIII and 3' XhoI restriction sites to facilitate subsequent cloning, was selected using the RNAit2 algorithm (https://dag.compbio.dundee.ac.uk/RNAit/) to ensure that the targeted sequence was distinct from any other genes to avoid any cross-RNAi effects [41]. This *FLAM8* DNA fragment was synthesized by GeneCust Europe (Dudelange, Luxembourg) and inserted into the HindIII-XhoI digested pZJM vector [39].

The pZJM-*FLAM8* plasmid was linearized with NotI prior to transfection using nucleofector technology (Lonza, Italy) as described previously [42]. The cell line was further engineered for endogenous tagging of *FLAM8* with an mNeonGreen (mNG) at its C-terminal end by using the p3329 plasmid [43], carrying a *FLAM8* gene fragment corresponding to *FLAM8* ORF nucleotides 8892–9391. Prior to nucleofection, NruI linearization of p3329-*FLAM8*-mNG plasmid was performed.

For *in vivo* experiments in mice, *FLAM8^RNAi* parasites were finally modified by integrating a plasmid encoding for the chimeric triple reporter which combines the red-shifted firefly luciferase PpyREH9 and the tdTomato red fluorescent protein fused with a TY1 tag [20]. Transformants were selected with the appropriate antibiotic concentrations: phleomycin (1 μg/mL), blasticidin (5 μg/mL), G418 (2 μg/mL), and puromycin (0.1 μg/mL). Clonal populations were obtained by limiting dilution. Cell culture growth was monitored with an automatic Muse cell analyzer (Merck Millipore, Paris).

## Generation of *FLAM8* KO mutants

For generating the *FLAM8* knockout and rescue cell lines, all insert templates were synthesized by GeneCust Europe (Dudelange, Luxembourg). For breaking the first allele, the 300 first nucleotides of the *FLAM8* gene flanking sequences were added on each side of a HYG resistance cassette (S2 Fig). For a complete disruption of the *FLAM8* locus, a second selectable marker (PAC) was flanked with the *FLAM8*-flanking sequence at 5' and by 300 nucleotides of the *FLAM8* ORF (nucleotides 501–800) at 3'. For generating an add-back rescue cell line, due to the large size of the *FLAM8* ORF (9,228 nucleotides), the PAC selection marker was replaced by a BLE marker cassette flanked by the 300 first nucleotides of the *FLAM8* 5' untranslated region (UTR) and by the nucleotides 1 to 500 of the *FLAM8* ORF for reinsertion into the endogenous locus of the knockout clone 1. PCR amplifications of the DNA fragments bearing the *FLAM8* flanking sequences and the appropriate resistance markers were used for nucleofection and generation of all cell lines. The primers used are listed below: 5'- CATGACTTTACGTGTTTGGGCAC-3' (FW, located 82 bp upstream the flanking 5'UTR sequence); 5'-CTTGCTTGTTTCTGTTTCGCAAC-3' (RV, 130 bp downstream the flanking 3'UTR sequence, used to replace one WT allele by HYG resistance cassette); 5'- GCACACTAAAACTCATTGAAAGCC-3' (RV, 926 bp downstream the ATG codon of *FLAM8*, used

**Table 1. Oligonucleotides used for PCR validation of the Δ*FLAM8* knockout and rescue cell lines.**

| Primer | Sequence | Target | |
|---|---|---|---|
| 1 | CATGACTTTACGTGTTTGGGCAC | *FLAM8* WT allele (82 nt upstream 5'UTR) | F |
| 2 | GCACACTAAAACTCATTGAAAGCC | *FLAM8* WT allele (926 nt downstream ATG) | R |
| 3 | CGTCCGAGGGCAAAGGAATAG | Hygromycin cassette | R |
| 4 | GACCGCGCACCTGGTGCATG | Puromycin cassette | R |
| 5 | GTGGCCGAGGAGCAGGACTGA | Phleomycin cassette | R |

Orientation of primers: F, forward; R, reverse.

for second WT allele replacement by PAC cassette and rescue line generation). All knockout and rescue cell lines were further transfected to express the chimeric triple reporter protein PpyRE9H/TY1/tdTomato for multimodal *in vivo* imaging approaches as described elsewhere [20]. Selection-marker recovery was confirmed by screening individual clones in multi-well plates. Transformants were selected with the appropriate antibiotic concentrations: phleomycin (1 μg/mL), blasticidin (5 μg/mL), puromycin (0.1 μg/mL) and hygromycin (2.5 μg/mL). Clonal populations were obtained by limiting dilution and cell culture growth was monitored with an automatic Muse cell analyzer (Merck Millipore, Paris).

Knockout and rescue cell lines were validated by whole-genome sequencing (BGI, Hong Kong). Briefly, genomic DNA from parental and mutant cell lines were sequenced by the HiSeq4000 sequence system (Illumina), generating about 10 million 100-bp reads and compared to that of the *T. brucei brucei* AnTat 1.1E Paris reference strain. In addition, some validation of the construct integrations in mutants were performed by PCR analysis according to standard protocols (S2 Fig and Table 1).

## Motility analyses

*In silico* 2D tracking was performed as previously described [44]. For each BSF strain, 10 to 20 movies were recorded for 20 seconds (50 ms of exposure). Trypanosomes at $1x10^6$ parasites/mL were maintained in matrix-dependent HMI-11 medium containing 0,5% or 1.1% methyl-cellulose at 37°C and were observed under the 10x objective of an inverted DMI-4000B microscope (Leica) coupled to an ORCA-03G (Hamamatsu) or a PRIME 95B (Photometrics) camera. Movies were converted with the MPEG Streamclip V.1.9b3 software (Squared 5) and analyzed with the medeaLAB CASA Tracking V.5.9 software (medea AV GmbH). Results were analyzed as mean ± SD of three independent experiments.

## *In vitro* bioluminescence quantification and analysis

To perform the parasite density / bioluminescence intensity assay, BSF parasites were counted, centrifuged, and resuspended in fresh HMI-11 medium. Then, 100 μL of this suspension containing $10^6$ parasites were transferred into black clear-bottom 96-well plates and serial 2-fold dilutions were performed in triplicate adjusting the final volume to 200 μL with 300 μg/mL of beetle luciferin (Promega, France). Luciferase activity was quantified after 10 min of incubation with an IVIS Spectrum imager (PerkinElmer). Imaging data analysis was performed with the Living Image 4.3.1 software (PerkinElmer) by drawing regions of interest with constant size for well plate quantification. Total photon flux was calculated after removal of intensity values from WT parasites and / or parasite-free medium corresponding to the background noise. Results were analyzed as mean ± SD of three independent experiments (S1 and S3A and S3B Figs).

## Mouse infection and ethical statements

Seven-week-old male BALB/c mice were purchased from Janvier Laboratory (sub-strain BALB/cAnNRj) and used as models for experimental infection and monitoring of the bioluminescence signal with the IVIS Spectrum imager (PerkinElmer). BR is authorized to perform experiments on vertebrate animals (license #A-75-2035) and is responsible for all the experiments conducted personally or under his supervision. For *in vivo* infections, groups of four and three animals (*FLAM8* knockdown and knockout-infected mice, respectively) were injected intraperitoneally (IP) with $10^5$ slender BSF parasites, washed in TDB (Trypanosome Dilution Buffer: 5 mM KCl, 80 mM NaCl, 1 mM $MgSO_4*7H_2O$, 20 mM $Na_2HPO_4$, 2 mM $NaH_2PO_4$, 20 mM glucose) and resuspended in 100 µl of PBS prior animal inoculation. To study the effects of the genotype factor (*FLAM8* knockdown) on parasitemia and EV parasite densities (variables), the most adapted statistical analysis was a multiple comparison of the means of 5 groups by ANOVA, using two-sided tests on the 5 experimental groups. With 3 individuals per group, mixed into 3 cages (5 individuals per cage) to limit any cage effect, the experimental design allows us to obtain significative results for strong effects superior to 1.13 with 0.05% confidence and a power fixed at 0.8. This experiment was performed in two independent replicates.

## *In vivo* bioluminescence imaging (BLI) analyses

Infection with bioluminescent parasites was monitored daily by detecting the bioluminescence signal in whole animals with the IVIS Spectrum imager (PerkinElmer). The equipment consists of a cooled charge-coupled camera mounted on a light-tight chamber with a nose cone delivery device to keep the mice anaesthetized during image acquisition with 1.5–2% isoflurane. A heated stage is comprised within the IVIS Spectrum imager to maintain optimum body temperature. D-luciferin potassium salt (Promega) stock solution was prepared in sterile PBS at 33.33 mg/mL and stored in a -20˚C freezer. To produce bioluminescence, mice were inoculated by the intraperitoneal route (IP) with 150 µL of D-luciferin stock solution (250 mg/Kg body weight). After 10 minutes of incubation to allow substrate dissemination, all mice were anaesthetized in an oxygen-rich induction chamber with 1.5–2% isoflurane, and dorsal and ventral BLI images were acquired by using automatic exposure (0.5 seconds to 5 minutes) depending on signal intensity.

Images were analyzed with the Living Image software version 4.3.1 (PerkinElmer). Data were expressed in total photons/second (p/s) corresponding to the total flux of bioluminescent signal according to the selected area (ventral and dorsal regions of interest with constant size covering the total body of the mouse). The background noise was removed by subtracting the bioluminescent signal of the control mouse from the infected ones for each acquisition. For parasite dissemination analyses, a minimum value of photons/second (p/s) was set for all animals in every time point to quantify the exact dissemination area (in $cm^2$) over the whole animal body. Parasitemia was determined daily following tail bleeds and assayed by automated fluorescent cell counting with a Muse cytometer (Merck-Millipore, detection limit at $10^2$ parasites/mL) according to the manufacturer's recommendations. The quantification of the total intravascular parasite population was assessed by calculating the blood volume of all animals according to their body weight and referring to daily parasitemia.

To quantify the total number of parasites by BLI (intravascular plus extravascular trypanosomes), an *in vivo* standard curve was performed (S3C and S3D Fig). Since the bioluminescent emission of cultured parental, KO subclones and rescue parasites was not significantly different (S3A and S3B Fig), the *in vivo* standard curve was obtained by injecting IP increasing numbers ($10^3$, $10^4$, $10^5$, $10^6$ and $10^7$ parasites/animal) of parental trypanosomes only (S3C and S3D Fig).

After 2.5h, animals received 150 μL of D-luciferin stock solution IP (250 mg/kg body weight), 10 minutes prior image acquisition. During this time, mice were anaesthetized with 1.5–2% iso-flurane, and images were acquired in the IVIS Spectrum imager by using automatic exposure settings. A region of interest with a constant size was used to correlate the number of injected parasites with the whole-body BLI signal. Non-infected controls were imaged and the total BLI values used to subtract the background signal or noise. The signals in photons/second were used to construct a standard curve to further interpolate the total number of trypanosomes present in each animal during the entire experimental infection period (S3D Fig). Subsequently, to obtain the number of extravascular parasites, the calculated total number of parasites present in the vascular system was subtracted from the total number of trypanosomes per animal body, resulting in estimating the total parasite population colonizing the extravascular compartments at a given time point.

### Endothelial transmigration assay

Single donor cryopreserved Primary Human Umbilical Vein Endothelial Cells (HUVECs) were obtained from Promocell and maintained as per the manufacturer's instructions in 75 cm$^2$ flasks at 37˚C with 5% $CO_2$ in endothelial cell growth medium (Promocell) with 100 U/mL penicillin, and 100 μg/mL streptomycin (Gibco). Cells were passaged at 80–90% confluence by dissociation with 0.04% Trypsin-0.03% EDTA (Promocell), split at 1:3–1:5 ratios, and maintained for up to six passages. Polyester transwell inserts for 24 well plates with 3 μm pore size (Corning) were coated with 10 μg/mL bovine fibronectin (Promocell) for one hour, the excess removed, and 600 μL pre-warmed endothelial cell growth medium added to the lower chamber (Fig 6A). The upper chamber was seeded with 2x10$^4$ HUVECs per insert in a volume of 100 μL endothelial cell growth medium. Media was exchanged in the upper and lower chamber every two days until monolayer confluence reached (approximately 6 days). Confluence was confirmed by FITC-dextran permeability assay and crystal violet staining of a sacrificed transwell. Briefly, 1 mg/mL FITC-70kDa dextran (Sigma-Aldrich) in endothelial cell growth medium was added to the upper chamber, incubated for 20 minutes and leakage to the lower chamber measured on a Qubit 4 Fluorometer (Invitrogen). Leakage of <1% of the FITC-dextran was observed at confluence. Monolayer integrity was confirmed by brightfield imaging of a transwell insert stained with crystal violet as per the manufacturer's instructions (Millipore). To perform the trypanosome transmigration assays, the confluent transwells were exchanged into a new 24 well plate containing pre-warmed assay media (endothelial cell growth medium supplemented with 20% trypanosome growth media) for two hours prior to performing the assay. Cultured trypanosomes were collected by centrifugation at 900×g for 5 min and resuspended in assay media at 2x10$^6$/mL. The media in the upper chamber was replaced with 100 μL of assay media containing 2x10$^5$ trypanosomes and incubated overnight at 37˚C, 5% $CO_2$. Each cell line was tested in triplicate transwells. Unbiased quantification of trypanosome transmigration was determined after 24 hours by collecting the media from the upper and lower chambers and transferring 100 μL to 96 well plates for automated counting using a Guava Easycyte HT system with a green laser and a custom counting protocol for tdTomato fluorescent trypanosomes. The cell counts per ml were used to calculate the number of trypanosomes in each compartment and proportion of transmigration into the lower chamber. Trypanosome transmigration was compared between the FLAM8 mutant cell lines and the parental reference cell line using the Generalised Linear Model function in R with a Gaussian family function and proportion transmigration as the dependant variable. A probability value of $p < 0.05$ was considered significant.

### *Ex vivo* RT-qPCR of mouse tissues

**Biological samples.**   After four weeks of *in vivo* monitoring, parental-, KO- and rescue-infected mice were euthanized by an overdose of an anesthetic/analgesic mixture of ketamine and xylazine (100 mg/kg and 10 mg/kg, respectively). Terminal bleeding was then performed through the inferior *vena cava*, and the collected blood was directly transferred in 1 ml RNAlater (ThermoFisher Scientific), snap frozen in liquid nitrogen for long-term preservation. Next, perfusion was initiated by injecting 0.9% NaCl (pre-heated at 37˚C), to completely remove the blood from the animal's body (both organs and vasculature), thus ensuring that subsequent analyses would only contain tissue-dwelling trypanosomes. Finally, spleen, lungs, kidney, gut, testis, liver and skin tissue samples were preserved in 1 ml RNAlater, snap frozen and stored in liquid nitrogen.

*RNA isolation*. Total RNA from samples was isolated using the QIAzol Reagent (Qiagen) according to the manufacturer's instructions. Frozen tissue samples were weighed and processed on ice to prevent thawing. Briefly, 30 mg of tissue were added to 700 µl of QIAzol lysis reagent, or 250 µl of blood in 550 µl of QIAzol lysis reagent, and samples were homogenized using the Precellys Evolution Homogenizer (Bertin, USA) with 2.8 mm stainless steel beads for 2 cycles of 1 min at 5000 rpm each, followed by 15 sec of resting between them. Homogenates were then incubated for 5 min at room temperature before the addition of 140 µl chloroform (0.2 volume of starting QIAzol lysis reagent), thoroughly mixed by vortexing for 15 seconds, incubated 5 min at room temperature and centrifuged at 12,000×g, 4˚C, 15 min. The aqueous phase containing the RNA was subsequently mixed with 1.5 volumes of absolute ethanol and transferred into a RNeasy Mini spin column (Qiagen) following the manufacturer's recommendations. The concentration of each RNA sample was measured by spectrophotometric analysis in a NanoDrop 2000c (ThermoFisher Scientific). Finally, RNA quality was determined by capillary electrophoresis in a 2100 Bioanalyzer (Agilent). Extracted RNA was stored at -80˚C prior to RT-qPCR analyses.

**DNAse treatment and validation.**   Extracted RNA samples were subjected to a second DNase treatment using Invitrogen's DNA-*free* kit (Life Technologies) according to manufacturer's protocol. DNase treatment confirmation was performed by running a qPCR targeting the *Tubulin* locus as housekeeping gene control. The primer pair used is as follows: forward (FW), 5'-ACTGGGCAAAGGGCCACTAC-3'; reverse (RV), 5'-CTCCTTGCAGCACA-CATCGA-3', with an amplicon size of 105 bp. Reactions were done in a volume of 20 µl containing: 1 µl of RNA template (17 ng), 10 µl of 2X GoTaq qPCR Master Mix Buffer, 2 µl of both FW and RV primers (at 10X) and 5 µl of Nuclease-Free Water. Amplification was accomplished in a QuantStudio 3 thermal cycler (Applied Biosystems) using the following program: 2 min at 95˚C; 40 cycles at 94˚C 15 sec; 55˚C 1 min; and final 30 sec at 60˚C. The absence of residual DNA in the DNAse-treated RNA samples was thus confirmed when no amplification was observed. RNA samples were further employed for gene expression quantification.

**Primer design of target genes.**   For *FLAM8* amplification, primers were designed to recognize the 324-bp region suppressed during *FLAM8* knockout generation: FW, 5'-GCATC GTTCGTGAGGTTGGA-3'; RV, 5'-GTTCCTCTTCGTCATCTGGTTCA-3'. The amplicon size was 88 bp. For Protein Associated to Differentiation 1 (PAD1) quantification in infected samples, primer sequences were described elsewhere [45] and are listed below: FW, 5'-T CATGGTTTCGCCATTCTCGTAACC-3'; RV, 5'-CTCAGCCACTTCTCTCCTACAAC AC-3'. Amplicon size is 156 bp.

**Real time RT-PCR assay.**   One step RT-PCR kit (Promega) was used to amplify *Tubulin*, *FLAM8* or *PAD1* targets (Table 2). Reactions were prepared in a volume of 20 µl, containing: 1 µl of RNA template (17 ng), 10 µl of 2X GoTaq qPCR Master Mix Buffer, 0.4 µl of 50X

**Table 2. Oligonucleotides used for *ex vivo* RT-qPCR of infected tissues.**

| Primer | | Sequence | Purpose |
|---|---|---|---|
| *Tubulin* | F | ACTGGGCAAAGGGCCACTAC | Housekeeping control. Total trypanosome quantification (both SL and ST parasites) through standard curve generation. |
| | R | CTCCTTGCAGCACACATCGA | |
| *FLAM8* | F | GCATCGTTCGTGAGGTTGGA | Detection of parasites expressing *FLAM8*. The amplified sequence is absent in *FLAM8* KO mutants. |
| | R | GTTCCTCTTCGTCATCTGGTTCA | |
| *PAD-1* | F | TCATGGTTTCGCCATTCTCGTAACC | Quantification of transmissible ST parasites within *ex vivo* samples. |
| | R | CTCAGCCACTTCTCTCCTACAACAC | |

Orientation of primers: F, forward; R, reverse.

GoScript RT Mix for 1-Step RT-qPCR, 2 μl of both FW and RV primers (10X), and 4,6 μl of Nuclease-Free Water. Reverse transcription and amplification were accomplished in one step in a QuantStudio 3 thermal cycler (Applied Biosystems) using the following incubation program: 15 min at 42°C; 10 min at 95°C; 40 cycles of 95°C during 30 sec; 55°C for *Tubulin*, 58°C for *FLAM8* or 60°C for *PAD1* during 1 min; final 72°C during 30 sec. A melt curve program was included: 15 sec at 95°C; 55°C for *Tubulin*, 58°C for *FLAM8* or 60°C for *PAD1* during 1 min; 95°C for 10 sec. Amplicons were then analyzed by gel electrophoresis. Negative and positive controls consisted of RNA extracted from uninfected mice and cultured trypanosomes, respectively.

**RT-qPCR data analysis.** All samples were amplified in triplicates and Cq mean values were calculated. Considering that the same initial amounts of total mRNAs extracted from each organ were used as RT-qPCR templates, the total number of parasites in each sample was calculated for comparisons by using a *Tubulin* RT-qPCR standard curve. Nine pools of cultured parasites (p) increasing by 10-folds from $10^1$ to $10^8$ were extracted and tested in triplicates by *Tubulin* RT-qPCR to generate a standard curve. The resulting standard curve's equation $Cq = -2,87 \times Log10(p) + 35,412$ allowed us to calculate the total number of parasites per mg of sample according to the Cq values obtained by *Tubulin* RT-qPCR on each sample. For normalization purposes to better compare the variations of the parasite populations in each compartment between strains, the difference between the number of parasites in each tissue sample of a given mouse and the number of parasites in the blood sample from the same mouse was calculated and plotted as Delta number of parasites. *Tubulin* expression (CqTub) was also used to normalize the *PAD1* mRNA levels (CqPAD1): the difference between the CqPAD1 and the CqTub values was calculated for each organ of each mouse and plotted as the Delta CqPAD1-CqTub. It allowed us to compare the relative proportions of parasites expressing *PAD1* mRNAs between organs and strains, a higher Delta CqPAD1-CqTub correlating with a lower amount of *PAD1* transcripts in the organ.

## Immunofluorescence analysis (IFA)

Cultured parasites were washed twice in TDB and spread directly onto poly-L-lysine coated slides. For methanol fixation, slides were air-dried for 10 min, fixed in methanol at -20°C for 30 seconds and rehydrated for 20 min in PBS. For immunodetection, slides were incubated for 1 h at 37°C with the appropriate dilution of the first antibody in 0.1% BSA in PBS. After 3 consecutive 5 min washes in PBS, species and subclass-specific secondary antibodies coupled to the appropriate fluorochrome (Alexa 488, Cy3, Cy5 Jackson ImmunoResearch) were diluted 1/400 in PBS containing 0.1% BSA and were applied for 1 h at 37°C. After washing in PBS as indicated above, slides were finally stained with 4',6-diamidino-2-phenylindole (DAPI, 1 μg/mL) for visualization of kinetoplast and nuclear DNA content and mounted under cover slips

with ProLong antifade reagent (Invitrogen), as previously described [8]. Slides were observed under an epifluorescence DMI4000 microscope (Leica) with a 100x objective (NA 1.4), an EL6000 (Leica) as light excitation source and controlled by the Micro-Manager V1.4.22 software (NIH), and images were acquired using an ORCA-03G (Hamamatsu) or a PRIME 95B (Photometrics) camera. Images were analyzed with ImageJ V1.8.0 (NIH). The monoclonal antibody mAb25 (anti-mouse IgG2a, 1:10) was used as a flagellum marker as it specifically recognizes the axoneme protein *Tb*SAXO1 [46]. FLAM8 was detected using: i) a specific rabbit serum (1:500) kindly provided by Paul McKean (University of Lancaster, UK), or ii) a monoclonal anti-mNeonGreen antibody (anti-mouse IgG2c, 1:100, ChromoTek). CARP3 was detected using a polyclonal CARP3 antiserum (1:150) [21]. Stumpy BSF were identified at the molecular level with a rabbit polyclonal anti-PAD1 antibody (kindly provided by Keith Matthews, University of Edinburgh; dilution 1:300) [47]. In the case of RNAi knockdown experiments, IFA signals were normalized using the signal obtained in non-induced controls as a reference.

### Measurements, normalization, and statistical analyses

Standardization of fluorescent signals was carried out by parallel setting of raw integrated density signals in all the images to be compared in ImageJ V1.8.0 (NIH). For clarity purposes, the brightness and contrast of several pictures were adjusted after their analysis in accordance with editorial policies. Statistical analyses and plots were performed with XLSTAT 2019.2.01 (Addinsoft) in Excel 2016 (Microsoft) or Prism V9.3.1 (GraphPad). Statistical analyses include: (1) linear regression for bioluminescence / fluorescence intensity vs. parasite density and RT-qPCR standard curve, (2) two-sided ANOVA tests with Tukey or Dunnett's ad-hoc post-tests for inter-group comparisons for growth curves, IV / EV parasite populations and parasite dissemination, and ΔCq comparisons of RT-qPCR data, all at 95% confidence.

### Supporting information

**S1 Fig. Validation of the triple-reporter efficiency in monomorphic *FLAM8^RNAi* parasite lines.** Linear correlation between the number of parasites and the bioluminescence (in p/s) emitted by monomorphic 427 *FLAM8::mNG FLAM8^RNAi* BSF overexpressing the triple reporter chimeric protein [20], acquired by the IVIS Spectrum imager. Parasites without the RNAi plasmid (control, C), non-treated with tetracycline (non-induced, NI) and treated with tetracycline (induced, I) are shown. Representative bioluminescent image of serial 1/2 dilutions performed in a 96-well plate (in photons / second / cm$^2$ / steradian). RNAi induction was triggered by the addition of 1 µg tetracycline and / or doxycycline for 72 h. Results represent the mean ± standard deviation (SD) of three independent experiments.
(EPS)

**S2 Fig. Molecular validation of the Δ*FLAM8* null mutant cell lines. A)** Whole-genome sequencing results showing *FLAM8* wild-type allele (WT, upper panel), the partial loss of the 5' *FLAM8* ORF in Δ*FLAM8* knockout trypanosomes (middle panel; the black arrow is showing the absence of reads at the *FLAM8* 5' ORF in the knockout line); and the restoration of the full *FLAM8* gene in rescue parasites (bottom panel; the black arrow is showing the absence of reads within the *pac* cassette due to the insertion of the new construct bearing the *ble* resistance marker), relative to the number of reads per 100-nt read length. The presence of the correct antibiotic cassettes is shown for knockout and rescue parasites (right middle and bottom panels). Δ*FLAM8* knockout and rescue parasites also bear a construct for expression of a triple reporter (TR) as assessed by the detection of *bsd* reads. **B)** Schemes showing the structure of

the *FLAM8* locus in wild-type parasites (upper scheme) and the integration plan of the different cassettes for Δ*FLAM8* knockout (middle panel, *HYG-* and *PAC*-containing schemes) and Rescue parasites (lower panel *HYG-* and *BLE*-containing schemes). **C)** PCR confirmation of the successful integrations of all reporter cassettes. Primer pairs used for PCRs are indicated at the bottom of each line and correspond to those drawn on the schemes in B), along with the expected band sizes of the corresponding diagnostic PCR. BSD: blasticidin; HYG: hygromycin; PAC: puromycin; BLE: phleomycin.
(EPS)

**S3 Fig. *In vitro* and *in vivo* validation of the triple-reporter efficiency in pleomorphic Δ*FLAM8* mutant cell lines. A)** Representative bioluminescent image (in photons / second) of serial 1/2 dilutions performed in a 96-well plate of pleomorphic parental, Δ*FLAM8* knockout subclones and rescue parasites overexpressing the triple reporter chimeric protein [20]. **B)** Linear correlation between the number of parasites and the emitted bioluminescence (in photons / second) acquired by the IVIS Spectrum imager. Results represent the mean ± standard deviation (SD) of three independent experiments. **C)** Representative ventral view images of mice infected with increasing amounts of parental trypanosomes ($10^3$, $10^4$, $10^5$, $10^6$ and $10^7$ parasites/animal) acquired with the IVIS Spectrum imager 2.5 hours after IP injection. **D)** *In vivo* standard curve showing the correlation between the number of injected parasites and the bioluminescent signal (in photons/second, $R^2 = 1$). The standard curve was further employed to calculate the total number of parasites present in infected animals through whole-body BLI signal.
(EPS)

**S4 Fig. The absence of *FLAM8* reduces extravascular trypanosome dissemination.** Detailed individual data used in Fig 3. Groups of BALB/c mice were injected IP with either one parental, three ΔF*LAM8* null subclones or one rescue strains. Total number of parasites in the blood of infected mice (intravascular, IV) daily counted from tail bleeds using a cytometer over 4 weeks (left graphics). Total number of extravascular trypanosomes quantified from bioluminescence images in the same mice (middle graphs). Dissemination measured over the entire animal body (in $cm^2$) through the total surface of bioluminescent signal (right graphs).
(EPS)

**S5 Fig. Functional investigations on the Δ*FLAM8* null mutants *in vivo* in the mammalian host.** Detailed individual data used in Fig 4. Groups of BALB/c mice were injected IP with either one parental, three ΔF*LAM8* null subclones or one rescue strains. Total number of parasites in the blood of infected mice (intravascular, IV) counted from tail bleeds using a cytometer over 3.5 weeks (left graphs). Total number of extravascular trypanosomes quantified from bioluminescence images in the same mice (middle graphs). Dissemination measured over the entire animal body (in $cm^2$) through the total surface of bioluminescent signal (right graphics).
(EPS)

**S6 Fig. Detailed individual data used in Fig 4D.** Delta number of parasites per dissected organs and strains in each individual mouse. The total number of parasites in each sample was calculated per mg of tissues by using a *Tubulin* RT-qPCR standard curve. **A)** For normalization purposes, the difference between the number of parasites in each tissue sample of a given mouse and the number of parasites in the blood sample from the same mouse was calculated and plotted as Delta number of parasites. **B)** As an alternative normalization method, the ratio between the number of parasites in each tissue sample of a given mouse and the number of parasites in the blood sample from the same mouse was calculated and plotted as EV / IV

parasites ratio.
(EPS)

**S1 Table. Detailed qPCR results from the second experimental infection used in Figs 4D, 4E, 5E and S6.**
(XLSX)

## Acknowledgments

We thank M. Bonhivers, D. Robinson, P. McKean, K. Matthews, and K. Gull for providing various plasmids and antibodies. We gratefully acknowledge the UTechS Photonic BioImaging (Imagopole), C2RT, Institut Pasteur, supported by the French National Research Agency (France BioImaging; ANR-10–INSB–04; Investments for the Future). We are grateful to P. Bastin for his strong scientific and human support. We warmly thank M. Boshart, P. Bastin, and S. Bonnefoy for their critical reading of the manuscript.

## Author Contributions

**Conceptualization:** Estefanía Calvo-Alvarez, Brice Rotureau.

**Data curation:** Estefanía Calvo-Alvarez, Jean Marc Tsagmo Ngoune, Anneli Cooper, Aïssata Camara.

**Formal analysis:** Estefanía Calvo-Alvarez, Jean Marc Tsagmo Ngoune, Parul Sharma, Anneli Cooper, Aïssata Camara.

**Funding acquisition:** Annette MacLeod, Brice Rotureau.

**Investigation:** Estefanía Calvo-Alvarez, Jean Marc Tsagmo Ngoune, Parul Sharma, Anneli Cooper, Aïssata Camara, Christelle Travaillé, Aline Crouzols.

**Methodology:** Estefanía Calvo-Alvarez, Jean Marc Tsagmo Ngoune, Anneli Cooper, Annette MacLeod, Brice Rotureau.

**Supervision:** Brice Rotureau.

**Writing – original draft:** Estefanía Calvo-Alvarez, Brice Rotureau.

**Writing – review & editing:** Estefanía Calvo-Alvarez, Jean Marc Tsagmo Ngoune, Parul Sharma, Anneli Cooper, Annette MacLeod, Brice Rotureau.

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
