## [Decision Letter · Decision Letter 0]

7 Apr 2023

Dear Dr Rotureau,

Thank you very much for submitting your manuscript "FLAgellum Member 8 modulates extravasation and extravascular distribution of African trypanosomes." for consideration at PLOS Pathogens. As with all papers reviewed by the journal, your manuscript was reviewed by members of the editorial board and by several independent reviewers. In light of the reviews (below this email), we would like to invite the resubmission of a significantly-revised version that takes into account the reviewers' comments.

Note, in particular, that all reviewers express shared concerns about substantial heterogeneity and robustness of extravasation results from mouse infections, questioning whether suitable numbers of replicates were used. This calls into question whether the main conclusion regarding extravasation is strongly supported and is crucial to address.

Reviewers 1 and 2 also both point out unexplained inconsistencies in the transmigration assays versus what would be expected, and reviewer 2 points out that these assays offer the opportunity to dissect the phenomena more fully, eg with more replicates and pairing of KO and rescue clones. 

It would be helpful to consider additional motility assays suggested by reviewer 3. 

Reviewers point out that some conclusions made regarding defects being due to signaling and regarding differential functions of FLAM8 in different life cycle stages are not directly addressed by the data and thus need to be tempered in the text. Solid suggestions for additional textual changes throughout the manuscript are provided by all reviewers.

We cannot make any decision about publication until we have seen the revised manuscript and your response to the reviewers' comments. Your revised manuscript is also likely to be sent to reviewers for further evaluation.

Sincerely,

Kent L. Hill

Academic Editor

PLOS Pathogens

James Collins III

Section Editor

PLOS Pathogens

Kasturi Haldar

Editor-in-Chief

PLOS Pathogens

orcid.org/0000-0001-5065-158X

Michael Malim

Editor-in-Chief

PLOS Pathogens

orcid.org/0000-0002-7699-2064

Reviewer's Responses to Questions

**Part I - Summary**

Reviewer #1: The study provides important insights into the function of FLAgellar Member 8 (FLAM8) in Trypanosoma brucei infection by demonstrating its role in mammalian tissue colonization. While FLAM8 is not essential for parasite survival and growth in vitro or stumpy differentiation, its absence resulted in “impaired” dissemination in the extravascular compartment and decreased trans-endothelial migration. The authors propose that FLAM8 functions as a docking platform for Cyclic AMP response protein 3 (CARP3), which plays a role in signalling, and proposed that environmental sensing and/or signalling may play a role in extravasation.

Overall, this study sheds light on the mechanisms involved in the colonization of tissue-dwelling T. brucei parasites in mammalian hosts. There are a few important issues that should be addressed.

Reviewer #2: This study by Alvarez et al. investigates the function of the protein FLAM8 in the protozoan parasite Trypanosoma brucei. Recent work from several groups has shown that during the insect stage in the parasite’s life cycle, FLAM8 is part of a complex at the tip of the parasite flagellum that includes adenylyl cyclase and CARP3 and disruption of this complex was shown to prevent colonization of the tsetse fly salivary glands. Intriguingly, FLAM8 localisation changes to a more dispersed flagellar signal in the bloodstream form that dwells in the bloodstream and tissues of the mammalian host. Based on this difference in localization this study was conducted to discover if FLAM8 has a distinct function in this stage.

The results show convincingly that FLAM8 is not essential for the survival of bloodstream forms in vitro or in vivo. The initial strategy of depleting the transcript by RNAi is a valid approach but since under knockdown conditions the cells maintained 40% of the transcript levels compared to controls, this does not allow for strong conclusions. More compelling evidence that FLAM8 is dispensable under the conditions tested is provided by the knockout strategy that was chosen next, which was to replace one allele completely and the second allele partially. The justification for this strategy was that it facilitated construction of rescue lines by adding back the missing portion. The validation of gene loss/truncation was done at the level of DNA and absence of RNA and protein expression was also tested. Overall, reasonable evidence is given that a functional null mutant was achieved. By all measures that were applied these mutant cells behaved the same in vitro as the control cells.

In vivo, in a mouse infection model, these FLAM8 mutants were able to establish a bloodstream infection and they were also found in extravascular sites. Overall these data show that bloodstream forms lacking FLAM8 remain infective, and the data even shows that they can differentiate to the stumpy forms capable of colonizing tsetse flies. These conclusions are well supported by the data.

The authors then analyse in more detail the proportion of parasites in extravascular spaces. The stated conclusion is that “FLAM8-null parasites exhibit a significantly impaired dissemination in the extravascular compartment, that is restored by the addition of a single rescue copy of FLAM8.”

While some of the data is compatible with this claim, there are major questions as to the strength of this data.

Reviewer #3: The role of flagellum in Trypanosoma brucei motility is well documented, but its role in environment sensing remains poorly explored. In this study, the authors investigated the phenotype of pleomorphic parasites lacking Flagellar member 8 during infection of the mammalian host. The authors discovered that FLAM8 ko parasites are defective in disseminating within the mammalian host. In vitro experiments indicate that this defect could be due to a deficiency in crossing the endothelial vessel wall. FLAM8 interacts with a cyclic AMP response protein 3 (CARP3) and this protein is displaced from the cell surface in the absence of FLAM8 is. The authors speculate that the dissemination defect could be due to poor sensing of the environment and deficient cAMP signaling.

The most important novelty of this work is the identification of the first parasite protein that shows a defect in tissue colonization. The depletion of this flagellar protein has little fitness impact in vitro, providing important knowledge that the parasite flagellum is crucial for parasites to extravasate into tissues.

The paper is well written and with a clear story line. However, more convincing show that the FLAM8 ko defect is robust and differentiation- and motility-independent, three major points need to be addressed experimentally.

**Part II – Major Issues: Key Experiments Required for Acceptance**

Reviewer #1: 1. The authors did a good job showing the variabilities that occur during T. brucei infection, especially by using three clones of the FLAM8 KO. However, there seems to be a fair amount of heterogeneity in the progression of infection that makes it hard to specifically say what the role of FLAM8 is during infection, and more directly what happens to the extravascular parasites, which is where the authors focus most of their work. In the first round of experiments, shown in Figure 3, there are significant differences in the EV parasites at day 7 onwards. In a repeat of this experiment, shown in Figure 4, it appears the EV parasites and dissemination measurements show that the FLAM8 KO achieves similar numbers after day 17. Considering the inconsistencies here, along with the smaller (but significant) differences seen at later stages of infection in the Fig 3 replicate, it seems that the simplest interpretation is that the FLAM8 KO is showing a delayed colonization of the extravascular compartments. The authors have used terms like “modulate, “contribute to the maintenance”, “involvement” to describe that FLAM8 is doing here, which is vague. The authors should clarify what they think is going on. It would also be better to show the individual data points that reflect the parasite numbers in each mouse, rather than just an average with error bars.

2. Along the same lines, can the authors provide some explanation for why FLAM KO clone 2 shows similar transmigration rates to the FLAM KO rescue? That clone had the lowest rate of EV parasites, at least by Fig 4D and 4E. It would seem that if there is some heterogeneity in infective capacity of the specific pleomorphs FLAM8 KO clones, the clone that retains near-wild type transmigration capacity should be better at entering the extravascular spaces.

3. The change in localization of FLAM8-CARP3 from the flagellar tip in PCFs to along the length of the flagellum in BSFs is very interesting, but there is nothing that supports the idea that this change in localization reflects a change in function. Both complexes are flagellar- there could be other reasons for why FLAM8-CARP3 change localization within the flagellum. The authors should show that that the change in localization leads to a change in function or tone down their interpretations at the end of the discussion (lines 353-372).

Reviewer #2: Major issues and additional data needed

1. Based on the data in Figures 3C and 3D the result is reported that dissemination / extravascular parasite numbers are lower for the mutants. Significant differences are noted for days 5-12 and again from day 19. If dissemination was impaired, what plausible explanation is there for the lack of a difference in the week between days 12-19?

The variation between the subclones is quite striking, and unexplained. The experiment was once repeated (Fig 4) and again, there was great variation between the clones in all tested conditions. The IV parasite data in Fig 4 A is particularly noisy, and different from the first experiment (Fig 3A). Given this variation, infecting only three mice with each cell line seems like a small number. This issue needs to be addressed e.g. by clarifying what calculations were done to assess that this was the appropriate number.

In Fig 4B and C initial differences are seen but the mutants all reach the same end point as the parental cell line. A much clearer difference is detected in Fig 4D where parasite burdens in isolated organs were quantified on day 24. However, this seems inconsistent with the data in Figs 4B and C that show NO difference on day 24.

2. In a transmigration assay with HUVEC cells a difference is reported in the transmigration between FLAM8 knockouts and parentals. The rescue cell line has an “intermediate” phenotype. The fact that it has only one allele is given as a possible explanation. However, the localization images in Fig 2 show no difference in FLAM8 signal between parental and rescue and no quantification is shown, leaving the question open how gene dosage relates to protein abundance and consequently phenotype. Moreover, the observation that the partial rescue did worse in the transmigration than one of the KOs and only slightly better than the other two KOs raises the question whether this measurement truly shows a rescue effect. It is also unclear which KO was the parent of the rescue. Considering the variation between KO clones, paired comparisons of KO clones with rescue lines derived from said KO are needed to allow for any conclusions.

3. The final experiment (Fig 6C) shows CARP3 was no longer detected in FLAM8 KO cells by immunofluorescence and it was detected at reduced levels in the rescue line. Images of only one cell each is presented as evidence. This should be backed up by a more quantitative measure, either from fluorescence measurements on a larger cell population or alternatively comparing protein levels e.g. on a Western blot. correlating the level of FLAM8 (using the antibody as in Fig 2) with the levels of CARP3 would also be important to make the link between the FLAM8 depletion phenotype and possible involvement of CARP3/cAMP signaling.

The claim that the phenotype is “possibly due to cAMP signaling impairments” is at this point speculative. There is no evidence in this paper that the defect reported for bloodstream forms has anything to do with cAMP.

Minor point: It is notable that Fig 6C shows not only lack of CARP3 signal in the flagellum but complete absence of any signal. Assuming the complete absence of CARP3 signal shown in Fig 6C is representative of the FLAM8 KO population, how is this interpreted? Do the authors conclude that absence of FLAM8 not only prevents CARP3 from localizing to the flagellum, but also leads to the degradation of CARP3 protein?

Overall my view is that the claims about a dissemination defect due to reduced extravasation are not strongly supported by the data. The effect is modest and temporary, in vivo. The number of biological repeats is small and the variation between mutant cell lines is large, both in vivo dissemination measures and in the transwell assay. The rescue is partial and some of the data is inconsistent as noted above.

The availability of the in vitro transmigration assay offers the opportunity to dissect the observed phenomena more fully. For example, the model proposed by the authors would predict that depletion of CARP3 would lead to a similar transmigration defect. This would be an important experiment.

Reviewer #3: The authors concluded that in FLAM8 knockout does not affect parasite motility in vitro, by measuring parasite speed and linearity in the presence of 0,5% methylcellulose. However, previous work by the Engstler lab (10.1371/journal.ppat.1005448) has shown that the type of motility displayed by T. brucei is highly dependent on the viscosity of the medium, which could reflect the viscosity of intra- and extravascular spaces. Therefore, it is crucial to demonstrate that FMAL8 knockout parasites have no motility defect at different concentrations of methylcellulose. Without these experiments or intravital imagining, the authors cannot exclude the possibility that FLAM8 knockout parasites have a motility defect in vivo.

The phenotype of parasite distribution in FLAM8 knockout in mice (Figures 3B-C) is not apparent due to many lines and many error bars that impede a clear interpretation of the graphs. Compared to wildtype parasites, FLAM knockout parasites appear to have lower numbers inside and outside vessels during the first week of infection. In later stages of the infection, FLAM8 knockout parasites appear to have higher numbers inside vessels and lower numbers outside vessels. It would be helpful if the authors could quantify the mean fold-difference between wildtype/knockout/AB cell lines at the different stages of infection and identify which differences are statistically significant. Moreover it would be interesting to explain the differences in phenotype in early and later stages of infection.

Assessing defects in differentiation in vivo is not trivial. The authors used PAD1 mRNA levels as a proxy for measuring the proportion of stumpy forms. It would be beneficial for the authors to perform additional assays such as cell cycle analysis by IFA (counting N and K) in tissue sections and/or morphology. Additionally, it is important to take into account the total parasite density in each organ and see if it is correlated with proportion of stumpy forms. Drawing conclusions about differences in the proportion of stumpy forms across organs (Figure 5F) based on just one time point of infection should be avoided. Figure 5G is adequate and compelling in demonstrating that FLAM8 ko does not appear to affect differentiation in vivo.

The connection between cAMP signaling and FLAM8 phenotype is solely based on changes in CARP3 localization, which is a weak link. Therefore, the speculation that FLAM8 phenotype may be due to sensing functions should be toned down.

**Part III – Minor Issues: Editorial and Data Presentation Modifications**

Reviewer #1: 1. Line 110: The authors may consider adding "life" to the sentence to clarify the protein's differential localization in different life cycle forms rather than cell cycle stages

2. Fig 5D: It would be helpful if the authors could explain how the ePCFs were characterized, whether based on morphology or the expression of a molecular marker.

3. Fig 6B: Could the authors comment on the variation observed in the different KO lines?

4. Fig 4 D-E: Could the authors explain the Y-axis in the figure legend to facilitate the comprehension of the graph by readers?

5. It would be interesting to investigate if expressing FLAM8 ectopically leads to more parasites migrating to the extravascular environment.

6. For the IF on CARP3 in Fig 6- are they using the same exposure/lookups? The CARP3 levels in the WT looks a lot lower than the add back. If there is a difference it would merit comment.

7. Figure 3B would benefit from a change to the Y-axis, perhaps with a 10^8 maximum, to spread out the data more. As currently plotted, it is very difficult to interpret day 5 to day 13.

8. Line 409: The website that the authors list for RNAit is no longer functional. There is a new one based out of Dundee for RNAit2.

Reviewer #2: (No Response)

Reviewer #3: To understand how intravascular parasites (Fig 1E) were counted, the reader has to check the Materials and methods section. Given that extravascular parasites (Fig 1F) are determined by subtracting this value from the number of parasites measured by total bioluminescence (Fig 1D), it would be important that the Results section or the legend of Figure 1 explains that intravascular parasites were counted from a tail bleed using a cytometer.

Is Figure 4A a replicate of Figure 3B? It sems completely different. How do authors explain such differences between experiments and what does it say about the FLAM8 phenotype?

The way in which the parasite densities are displayed in the graphs of Figure 4D-E is not very user-friendly. It would be beneficial for readers to not only see the delta, but also be aware of the total number of parasites for each organ and the ratio delta/total, which indicates the proportion of parasite population is missing in a given organ upon FLM8 knockout. Currently, the way the data is plotted makes it difficult to comprehend the relative effect size. Additionally, error bars are missing in panel D, and legend of Y axis should be more precise in explaining what Delta represents. Finally, the figure legend should indicate the day of infection when the organs were collected.

The way the proportion of stumpy forms is displayed in Figure 5F-G is confusing. A value of 3 to 9 in Delta is not intuitive.

The title of the paper is confusing and long. Would the authors consider the alternative title “FLAgellum Member 8 modulates extravascular distribution of African trypanosomes.”

PLOS authors have the option to publish the peer review history of their article (what does this mean?). If published, this will include your full peer review and any attached files.

Reviewer #1: No

Reviewer #2: No

Reviewer #3: **Yes: **Luisa M Figueiredo
---

## [Decision Letter · Decision Letter 1]

22 Nov 2023

Dear Dr Rotureau,

Thank you very much for submitting your manuscript "FLAgellum Member 8 modulates extravascular distribution of African trypanosomes." for consideration at PLOS Pathogens. As with all papers reviewed by the journal, your manuscript was reviewed by members of the editorial board and by several independent reviewers. The reviewers appreciated the attention to an important topic, though they differed with respect to their evaluation of the revised work. Upon thorough consideration of reviewer comments and the revised paper, we are willing to accept this manuscript for publication, providing that you modify the text to address the concern below, and this will be evaluated by editorial staff without the need to send again to external reviewers.

Provide some additional comment on heterogeneity among FLAM8KO clones, e.g. Fig 6A result that addback does not fully rescue. Incomplete rescue is not, per se, a major problem, as it is recognized that addback of a single allele combined with a complex phenotype and technologically challenging assay may explain the heterogeneity among clones. However, more comment on heterogeneity among KO clones, particularly vs addback in Fig 6A, and acknowledgment of the relatively noisy data in some cases (e.g. per reviewer 1 comments about Fig 3 and 4) needs to be made. Currently the results and Discussion sections tend to gloss over this.

Sincerely,

Kent L. Hill

Academic Editor

PLOS Pathogens

James Collins III

Section Editor

PLOS Pathogens

Kasturi Haldar

Editor-in-Chief

PLOS Pathogens

orcid.org/0000-0001-5065-158X

Michael Malim

Editor-in-Chief

PLOS Pathogens

orcid.org/0000-0002-7699-2064

Reviewer Comments (if any, and for reference):

Reviewer's Responses to Questions

**Part I - Summary**

Reviewer #1: (No Response)

Reviewer #3: The authors have addressed my main concerns. Some of the proposed experiments/analysis were undertaken.

In my opinion, Figures 3D-E still appear to show that the phenotype of FLAM8 is different early and late in infection. But a more robust analysis using windows of 3-4 days would probably be more revealing than a daily statistical comparison.

Although the mechanism remains unknown, these findings open the door to understanding how T. brucei parasites extravasate into tissues.

**Part II – Major Issues: Key Experiments Required for Acceptance**

Reviewer #1: While the revised manuscript has addressed some of the reviewer comments, I still have reservations about the reproducibility of the key data in Figure 3 and 4. The decline in the EV populations in the FLAM8 KO are fairly small, at least how they are depicted in Fig 3E. It seems like the normalization method, such at the dissemination shown in Fig 3F (comparing time points 19-27), can greatly influence the magnitude and significance of the differences in parasite number, which makes it hard to know which is an accurate representation of the results.

This is even more concerning in Figure 4, where many of the significant differences that were seen in Fig 3 don't appear to reproduce. I realize that in vivo infections come with a fair amount of heterogeneity, but it feels that we have two experiments here that are saying different things, with the interpretation heavily leaning on only one outcome. If the results in Fig 3 and Fig 4 are given equal weight, then the FLAM8 KO appears to impact EV dissemination early in the infection, but not during later stages. The additional confounder here is that the qPCR data in Figs 4D-E appears to conflict with the data Fig 4B-C, showing fairly clear differences in parasite numbers (are they statistically significant?) during the late stages of the infection that are not captured with the whole animal imaging. This brings up the possibility that there are significant differences between WT and the FLAM8 KO EV dissemination, but at the expense of suggesting that the whole-animal imaging approach is not reliable/sensitive enough to pick up the differences in parasite numbers in this implementation. If this is the case, I don't know how to interpret the results in the paper.

Reviewer #3: (No Response)

**Part III – Minor Issues: Editorial and Data Presentation Modifications**

Reviewer #1: In Figure 5E, I find it strange to average the parasite levels for different organs, considering differences in size and what they can tolerate in terms of parasite burden. Is this the only way to get statistics for the organ parasite numbers?

Reviewer #3: (No Response)

PLOS authors have the option to publish the peer review history of their article (what does this mean?). If published, this will include your full peer review and any attached files.

Reviewer #1: No

Reviewer #3: **Yes: **Luisa M Figueiredo

Figure Files:

Data Requirements:

Reproducibility:

References:

---

## [Editor Report · Decision Letter 2]

6 Dec 2023

Dear Dr Rotureau,

We are pleased to inform you that your manuscript 'FLAgellum Member 8 modulates extravascular distribution of African trypanosomes.' has been provisionally accepted for publication in PLOS Pathogens.

Best regards,

Kent L. Hill

Academic Editor

PLOS Pathogens

James Collins III

Section Editor

PLOS Pathogens

Kasturi Haldar

Editor-in-Chief

PLOS Pathogens

orcid.org/0000-0001-5065-158X

Michael Malim

Editor-in-Chief

PLOS Pathogens

orcid.org/0000-0002-7699-2064
---

## [Editor Report · Acceptance letter]

15 Dec 2023

Dear Dr Rotureau,

We are delighted to inform you that your manuscript, "FLAgellum Member 8 modulates extravascular distribution of African trypanosomes.," has been formally accepted for publication in PLOS Pathogens.

Best regards,

Michael Malim

Editor-in-Chief

PLOS Pathogens

orcid.org/0000-0002-7699-2064